# THE EFFECTIVE HORIZON EXPLAINS DEEP RL PERFORMANCE IN STOCHASTIC ENVIRONMENTS

**Cassidy Laidlaw**  **Banghua Zhu**  **Stuart Russell**  **Anca Dragan**
University of California, Berkeley
`{cassidy_laidlaw,banghua,russell,anca}@cs.berkeley.edu`

## ABSTRACT

Reinforcement learning (RL) theory has largely focused on proving minimax sample complexity bounds. These require *strategic* exploration algorithms that use relatively limited function classes for representing the policy or value function. Our goal is to explain why deep RL algorithms often perform well in practice, despite using *random* exploration and much more expressive function classes like neural networks. Our work arrives at an explanation by showing that many stochastic MDPs can be solved by performing only a few steps of value iteration on the random policy's Q function and then acting greedily. When this is true, we find that it is possible to separate the *exploration* and *learning* components of RL, making it much easier to analyze. We introduce a new RL algorithm, SQIRL, that iteratively learns a near-optimal policy by exploring randomly to collect rollouts and then performing a limited number of steps of fitted-Q iteration over those rollouts. We find that any regression algorithm that satisfies basic in-distribution generalization properties can be used in SQIRL to efficiently solve common MDPs. This can explain why deep RL works with complex function approximators like neural networks, since it is empirically established that neural networks generalize well in-distribution. Furthermore, SQIRL explains why random exploration works well in practice, since we show many environments can be solved by effectively estimating the random policy's Q-function and then applying zero or a few steps of value iteration. We leverage SQIRL to derive instance-dependent sample complexity bounds for RL that are exponential only in an "effective horizon" of lookahead—which is typically much smaller than the full horizon—and on the complexity of the class used for function approximation. Empirically, we also find that SQIRL performance strongly correlates with PPO and DQN performance in a variety of stochastic environments, supporting that our theoretical analysis is predictive of practical performance. Our code and data are available at `https://github.com/cassidylaidlaw/effective-horizon`.

## 1 INTRODUCTION

The theory of reinforcement learning (RL) does not quite predict the practical successes (and failures) of deep RL. Specifically, there are two gaps between theory and practice. First, whereas RL theory emphasizes strategic exploration, practical deep RL algorithms often resort to random exploration, such as $\epsilon$-greedy strategies. This divergence is difficult to resolve since theory predicts exponential worst-case sample complexity for random exploration. Most recent progress in the theory of RL has focused on strategic exploration algorithms, which use upper confidence bound (UCB) bonuses to effectively explore the state space of an environment. Second, RL theory struggles to incorporate complex function approximators like the neural networks used in deep RL. UCB-type algorithms only work in highly-structured environments where they can use simple function classes to represent value functions and policies.

Our goal is to bridge these two gaps: to explain why random exploration works despite being exponentially bad in the worst-case, and to understand why deep RL succeeds despite using deep neural networks for function approximation. Some recent progress has been made on the former problem by Laidlaw et al. (2023), who analyze when random exploration will succeed in *deterministic* environments. Their analysis begins by demonstrating a surprising property: in many determinis-

tic environments, it is optimal to act greedily according to the Q-function of the policy that takes actions uniformly at random. This inspires their definition of a property of deterministic environments called the "effective horizon," which is roughly the number of lookahead steps a Monte Carlo planning algorithm needs to solve the environment when relying on random rollouts to evaluate leaf nodes. They then show that a randomly exploring RL algorithm called Greedy Over Random Policy (GORP) has sample complexity exponential only in the effective horizon rather than the full horizon. While the effective horizon is sometimes equal to the full horizon, they show it is much smaller for many benchmark environments where deep RL succeeds; conversely, when the effective horizon is high, deep RL rarely works.

In this work, we take inspiration from the effective horizon to analyze RL in stochastic environments with function approximation. A major challenge of understanding RL in this setting is the complex interplay between exploration and function approximation. This has made strategic exploration algorithms based on upper confidence bound (UCB) bonuses difficult to analyze because the bonuses must be carefully propagated through the function approximators. The same issue makes it hard to understand deep RL algorithms, in which the current exploration policy affects the data the function approximators are trained on, which in turn affects future exploration. Our idea is to leverage the effective horizon assumption—that limited lookahead followed by random rollouts is enough to arrive at the optimal action—to separate exploration and learning in RL.

We introduce a new RL algorithm, SQIRL (shallow Q-iteration via reinforcement learning), that generalizes GORP to stochastic environments. SQIRL iteratively learns a policy by alternating between collecting data through purely random exploration and then training function approximators on the collected data. During the training phase, SQIRL uses *regression* to estimate the random policy's Q-function and then *fitted Q-iteration* to approximate a few steps of value iteration. The advantage of this algorithm is that it only relies on access to a *regression oracle* that can generalize in-distribution from i.i.d. samples—a property that is empirically true of neural networks. Thus, unlike strategic exploration algorithms which work for only limited function classes, SQIRL helps explain why RL can work with expressive function classes. Furthermore, the way SQIRL leverages the effective horizon property helps explain why RL works in practice using random exploration.

Theoretically, we prove instance-dependent sample complexity bounds for SQIRL that depend on a stochastic version of the effective horizon as well as properties of the regression oracle used. We demonstrate empirically that the effective horizon assumptions are satisfied in many stochastic benchmark environments. Furthermore, we show that a wide variety of function approximators can be used within SQIRL. For instance, our bounds hold for least-squares regression with function classes of finite pseudo-dimension, including linear functions, neural networks, and many others.

To strengthen our claim that SQIRL can often explain why deep RL succeeds while using random exploration and neural networks, we compare its performance to PPO (Schulman et al., 2017) and DQN (Mnih et al., 2015) in over 150 stochastic environments. We implement SQIRL using least-squares neural network regression and evaluate its empirical sample complexity, along with that of PPO and DQN, in sticky-action versions of the BRIDGE environments from Laidlaw et al. (2023). We find that in environments where both PPO and DQN converge to an optimal policy, SQIRL does as well 85% of the time; when both PPO and DQN fail, SQIRL never succeeds. The strong performance of SQIRL in these stochastic environments implies both that the effective horizon of most of the environments is low and that our regression oracle assumption is met by the neural networks used in SQIRL. Furthermore, the strong relationship between the performance of SQIRL and that of deep RL algorithms suggests that deep RL generally succeeds using the same properties.

These empirical results, combined with our theoretical contributions, show that the effective horizon and the SQIRL algorithm can help explain when and why deep RL works even in stochastic environments. There are still some environments in our experiments where SQIRL fails while PPO or DQN succeeds, suggesting lines of inquiry for future research to address. However, we find that SQIRL's performance is as similar to PPO and DQN as their performance is to each other's, suggesting that SQIRL and the effective horizon explain a significant amount of deep RL's performance.

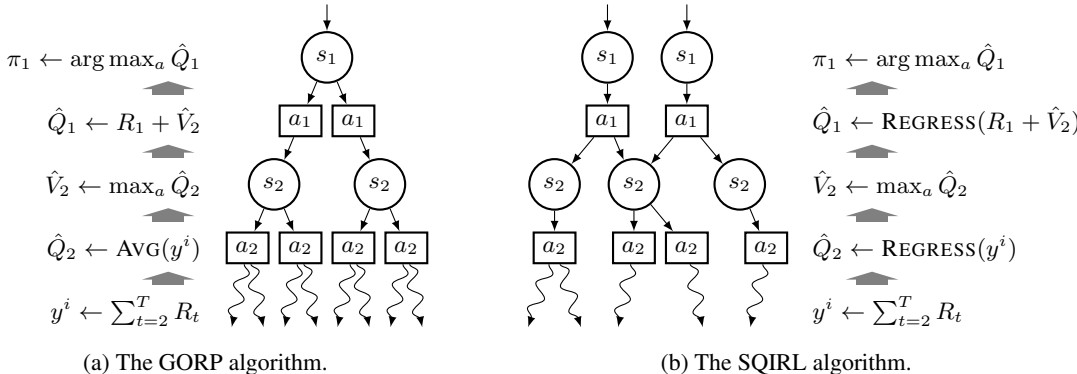

(a) The GORP algorithm.

(b) The SQIRL algorithm.

Figure 1: We introduce the shallow Q-iteration via reinforcement learning (SQIRL) algorithm, which uses random exploration and function approximation to efficiently solve environments with a low stochastic effective horizon. SQIRL is a generalization of the GORP algorithm (Laidlaw et al., 2023) to stochastic environments. In the figure, both algorithms are shown solving the first timestep of a 2-QVI-solvable MDP. The GORP algorithm (left) uses random rollouts to estimate the random policy's Q-values at the leaf nodes of a "search tree" and then backs up these values to the root node. It is challenging to generalize this algorithm to stochastic environments because both the initial state and transition dynamics are random. This makes it impossible to perform the steps of GORP where it averages over random rollouts and backs up values along deterministic transitions. SQIRL replaces these steps with *regression* of the random policy's Q-values at leaf nodes and *fitted Q-iteration* (FQI) for backing up values, allowing it to efficiently learn in stochastic environments.

## 2 SETUP AND RELATED WORK

We consider the setting of an episodic Markov decision process (MDP) with finite horizon. The MDP comprises a horizon $T \in \mathbb{N}$, states $s \in \mathcal{S}$, actions $a \in \mathcal{A}$, initial state distribution $p_1(s_1)$, transitions $p_t(s_{t+1} \mid s_t, a_t)$, and reward $R_t(s_t, a_t)$ for $t \in [T]$, where $[n]$ denotes the set $\{1, \ldots, n\}$. We assume that $A = |\mathcal{A}| \geq 2$ is finite. While we do not explicitly consider discounted MDPs, our analysis is easily extendable to incorporate a discount rate.

An RL agent interacts with the MDP for a number of episodes, starting from a state $s_1 \sim p(s_1)$. At each step $t \in [T]$ of an episode, the agent observes the state $s_t$, picks an action $a_t$, receives reward $R(s_t, a_t)$, and transitions to the next state $s_{t+1} \sim p(s_{t+1}, s_t, a_t)$. A policy $\pi$ is a set of functions $\pi_1, \ldots, \pi_t : \mathcal{S} \to \Delta(\mathcal{A})$, which defines for each state and timestep a distribution $\pi_t(a \mid s)$ over actions. If a policy is deterministic at some state, then with slight abuse of notation we denote $a = \pi_t(s)$ to be the action taken by $\pi_t$ in state $s$. We assume that the total reward $\sum_{t=1}^{t} R_t(s_t, a_t)$ is bounded almost surely in $[0, 1]$; any bounded reward function can be normalized to satisfy this assumption.

Using a policy to select actions in an MDP induces a distribution over states and actions with $a_t \sim \pi_t(\cdot \mid s_t)$. We use $P_\pi$ and $E_\pi$ to refer to the probability measure and expectation with respect to this distribution for a particular policy $\pi$. We denote a policy's Q-function $Q_t^\pi : \mathcal{S} \times \mathcal{A} \to \mathbb{R}$ and value function $V_t^\pi : \mathcal{S} \to \mathbb{R}$ for each $t \in [T]$, defined as:

$$x^{'}Q_t^\pi(s, a) = E_\pi \left[ \sum_{t'=t}^{T} R_{t'}(s_{t'}, a_{t'}) \mid s_t = s, a_t = a \right] \quad V_t^\pi(s) = E_\pi \left[ \sum_{t'=t}^{T} R_{t'}(s_{t'}, a_{t'}) \mid s_t = s \right]$$

Let $J(\pi) = E_{s_1 \sim p(s_1)}[V_1^\pi(s_1)]$ denote the expected return of a policy $\pi$. The objective of an RL algorithm is to find an $\epsilon$-optimal policy, i.e., one such that $J(\pi) \geq J^* - \epsilon$ where $J^* = \max_{\pi^*} J(\pi^*)$.

Suppose that after interacting with the environment for $n$ timesteps (i.e., counting one episode as $T$ timesteps), an RL algorithm returns a policy $\pi^n$. We define the $(\epsilon, \delta)$ sample complexity $N_{\epsilon, \delta}$ of an RL algorithm as the minimum number of timesteps needed to return an $\epsilon$-optimal policy with probability at least $1 - \delta$, where the randomness is over the environment and the RL algorithm:

$$N_{\epsilon, \delta} = \min \left\{ n \in \mathbb{N} \mid \mathbb{P} \left( J(\pi^n) \geq J^* - \epsilon \right) \geq 1 - \delta \right\}.$$

### 2.1 RELATED WORK

As discussed in the introduction, most prior work in RL theory has focused finding strategic exploration-based RL algorithms which have minimax regret or sample complexity bounds (Jiang et al., 2017; Azar et al., 2017; Jin et al., 2018; 2019; Sun et al., 2019; Yang et al., 2020; Dong et al.,

2020; Domingues et al., 2021), i.e., they perform well in worst-case environments. However, since the worst-case bounds for random exploration are exponential in the horizon (Koenig & Simmons, 1993; Jin et al., 2018), minimax analysis cannot explain why random exploration works well in practice. Furthermore, while strategic exploration has been extended to broader and broader classes of function approximators (Jin et al., 2021; Du et al., 2021; Foster et al., 2021; Chen et al., 2022), even the broadest of these still requires significant linear or low-rank structure in the environment. This also limits the ability of strategic exploration analysis to explain or improve on practical deep RL algorithms that use neural networks to succeed in unstructured environments. While some work has theoretically analyzed random exploration (Liu & Brunskill, 2019; Dann et al., 2022) and more general function approximators (Malik et al., 2021) in RL, Laidlaw et al. (2023) show that the sample complexity bounds in these papers fail to explain empirical RL performance even in deterministic environments.

The SQIRL algorithm is partially inspired by fitted Q-iteration (FQI) (Ernst et al., 2005) and previous analyses of error propagation in FQI (Antos et al., 2007; Munos & Szepesvári, 2008). While FQI has been analyzed for planning (Hallak et al., 2023) or model-based RL (Argenson & Dulac-Arnold, 2021), our analysis is novel because it uses FQI in a model-free RL algorithm which leverages the effective horizon assumption to perform well in realistic environments. SQIRL is also related to *lookahead*, which Bertsekas (2020; 2022) suggests often quickly converges to an optimal policy because it is equivalent to a step of Newton's method for finding a fixed-point of the Bellman equation (Kleinman, 1968; Puterman & Brumelle, 1979). However, lookahead is mainly used in model-based settings and/or deterministic environments. By showing that a model-free algorithm (SQIRL) can approximate lookahead, we find that similar principles underly the success of model-free deep RL.

## 3  THE STOCHASTIC EFFECTIVE HORIZON AND SQIRL

We now present our main theoretical findings extending the effective horizon and GORP algorithm to stochastic environments. The effective horizon was motivated in Laidlaw et al. (2023) by a surprising property that holds in many deterministic MDPs: acting greedily with respect to the Q-function of the random policy, i.e. $\pi^{\mathrm{rand}}{}_t(a \mid s) = 1/A \quad \forall s, a, t$, gives an optimal policy. Even when this property doesn't hold, the authors find that applying a few steps of value iteration to the random policy's Q-function and then acting greedily is often optimal; they call this property *k-QVI-solvability*. We begin by investigating whether this property holds in common stochastic environments.

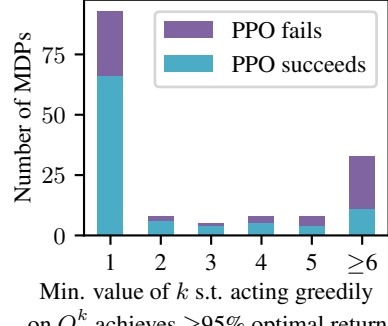

Min. value of $k$ s.t. acting greedily on $Q^k$ achieves $\geq 95\%$ optimal return

Figure 2: Among sticky-action versions of the MDPs in the BRIDGE dataset, more than half can be approximately solved by acting greedily with respect to the random policy's Q-function ($k = 1$); many more can be by applying just a few steps of Q-value iteration before acting greedily ($2 \leq k \leq 5$). When $k$ is low, we observe that deep RL algorithms like PPO are much more likely to solve the environment.

To define $k$-QVI-solvability, we introduce some notation. One step of Q-value iteration transforms a Q-function $Q$ to $Q' = \mathrm{QVI}(Q)$, where

$$Q'_t(s_t, a_t) = R_t(s_t, a_t) + E_{s_{t+1}} \left[\max_{a \in \mathcal{A}} Q_{t+1}(s_{t+1}, a)\right].$$

We also denote by $\Pi(Q)$ the set of policies which act greedily with respect to the Q-function $Q$; that is,

$$\Pi(Q) = \left\{\pi \ \middle| \ \forall s, t \quad \pi_t(s) \in \arg\max_{a \in \mathcal{A}} Q_t(s, a)\right\}.$$

Furthermore, we define a sequence of Q-functions $Q^1, \ldots, Q^T$ by letting $Q^1 = Q^{\pi^{\mathrm{rand}}}$ be the Q-function of the random policy and $Q^{i+1} = \mathrm{QVI}(Q^i)$.

**Definition 3.1** (*k-QVI-solvable*). *We say an MDP is $k$-QVI-solvable for some $k \in [T]$ if every policy in $\Pi(Q^k)$ is optimal.*

If acting greedily on the random policy's Q-values is optimal, then an MDP is 1-QVI-solvable; $k$-QVI-solvability extends this to cases where value iteration must be applied to the Q-function first.

To see if stochastic environments are commonly $k$-QVI-solvable for small values of $k$, we constructed sticky-action versions of the 155 deterministic MDPs in the BRIDGE dataset (Laidlaw et al., 2023). Sticky actions are a common and effective method for turning deterministic MDPs into

stochastic ones (Machado et al., 2018) by introducing a 25% chance at each timestep of repeating the action from the previous timestep. We analyzed the minimum values of $k$ for which these MDPs are approximately $k$-QVI-solvable, i.e., where one can achieve at least 95% of the optimal return (measured from the minimum return) by acting greedily with respect to $Q^k$. The results are shown in Figure 2. Many environments are approximately $k$-QVI-solvable for very low values of $k$; more than half are approximately 1-QVI-solvable. Furthermore, these are the environments where deep RL algorithms like PPO are most likely to find an optimal policy, suggesting that $k$-QVI-solvability is key to deep RL's success in stochastic environments.

While many of the sticky-action stochastic MDPs created from the BRIDGE dataset are $k$-QVI-solvable for small $k$, this alone is not enough to guarantee that random exploration can lead to efficient RL. Laidlaw et al. (2023) define the effective horizon by combining $k$ with a measure of how precisely $Q^k$ needs to be estimated to act optimally.

**Definition 3.2** ($k$-gap). *If an MDP is $k$-QVI-solvable, we define its $k$-gap as*
$$\Delta_k = \inf_{(t,s) \in [T] \times \mathcal{S}} \left( \max_{a^* \in \mathcal{A}} Q_t^k(s, a^*) - \max_{a \notin \arg\max_a Q_t^k(s,a)} Q_t^k(s, a) \right).$$

Intuitively, the smaller the $k$-gap, the more precisely an algorithm must estimate $Q^k$ in order to act optimally in an MDP which is $k$-QVI-solvable. We can now define the *stochastic* effective horizon, which we show is closely related to the effective horizon in deterministic environments:

**Definition 3.3** (Stochastic effective horizon). *Given $k \in [T]$, define $\bar{H}_k = k + \log_A(1/\Delta_k^2)$ if an MDP is $k$-QVI-solvable and otherwise $\bar{H}_k = \infty$. The stochastic effective horizon is $\bar{H} = \min_k \bar{H}_k$.*

**Lemma 3.4.** *The deterministic effective horizon $H$ is bounded as*
$$H \le \min_k \left[ \bar{H}_k + \log_A O \left( \log \left( T A^k \right) \right) \right].$$
*Furthermore, if an MDP is $k$-QVI-solvable, then with probability at least $1 - \delta$, GORP will return an optimal policy with sample complexity at most $O(kT^2 A^{\bar{H}_k} \log(TA/\delta))$.*

We defer all proofs to the appendix. Lemma 3.4 shows that our definition of the stochastic effective horizon is closely related to the deterministic effective horizon definition: it is an upper-bound up to logarithmic factors. Furthermore, it can bound the sample complexity of the GORP algorithm in deterministic environments. The advantage of the stochastic effective horizon definition is that it does not rely on the GORP algorithm, but is rather defined based on basic properties of the MDP; thus, it equally applies to stochastic environments. However, it is still unclear how a low effective horizon can lead to provably efficient RL in stochastic MDPs.

## 3.1 SQIRL

To show that the stochastic effective horizon can provide insight into when and why deep RL succeeds, we introduce the shallow Q-iteration via reinforcement learning (SQIRL) algorithm. Recall the two theory-practice divides we aim to bridge: first, understanding why random exploration works in practice despite being exponentially inefficient in theory; and second, explaining why using deep neural networks for function approximation is feasible in practice despite having little theoretical justification. SQIRL is designed to address both of these. It generalizes the GORP algorithm to stochastic environments, giving sample complexity exponential only in the stochastic effective horizon $\bar{H}$ rather than the full horizon $T$. It also allows the use of a wide variety of function approximators that only need to satisfy relatively mild conditions; these are satisfied by neural networks and many other function classes.

---

**Algorithm 1** The greedy over random policy (GORP) algorithm, used to define the effective horizon in deterministic environments.

---

1: **procedure** GORP($k, m$)
2:     **for** $i = 1, \dots, T$ **do**
3:         **for** $a_{i:i+k-1} \in \mathcal{A}^k$ **do**
4:             sample $m$ episodes following $\pi_1, \dots, \pi_{i-1}$, then actions $a_{i:i+k-1}$, and finally $\pi^{\text{rand}}$.
5:             $\hat{Q}_i(s_i, a_{i:i+k-1}) \leftarrow$
                $\frac{1}{m} \sum_{j=1}^m \sum_{t=i}^T \gamma^{t-i} R(s_t^j, a_t^j)$.
6:         **end for**
7:         $\pi_i(s_i) \leftarrow \arg\max_{a_i \in \mathcal{A}}$
            $\max_{a_{i+1:i+k-1} \in \mathcal{A}^{k-1}} \hat{Q}_i(s_i, a_i, a_{i+1:i+k-1})$.
8:     **end for**
9:     **return** $\pi$
10: **end procedure**

---

**GORP** The GORP algorithm (Algorithm 1 and Figure 1a) is difficult to generalize to the stochastic case because many of its components are specific to deterministic environments. GORP learns

---

**Algorithm 2** The shallow Q-iteration via reinforcement learning (SQIRL) algorithm.

1: **procedure** SQIRL($k, m$, REGRESS)
2:     **for** $i = 1, \ldots, T$ **do**
3:         Collect $m$ episodes by following $\pi_t$ for $t < i$ and $\pi^{\text{rand}}$ thereafter to obtain $\{(s_t^j, a_t^j, y_t^j)\}_{j=1}^m$.
4:         $\hat{Q}_{i+k-1}^1 \leftarrow \text{REGRESS}(\{(s_{i+k-1}^j, a_{i+k-1}^j, \sum_{t=i+k-1}^T R_t(s_t^j, a_t^j))\}_{j=1}^m)$.
5:         **for** $t = i + k - 2, \ldots, i$ **do**
6:             $\hat{Q}_t^{i+k-t} \leftarrow \text{REGRESS}(\{(s_t^j, a_t^j, R_t(s_t^j, a_t^j) + \max_{a \in \mathcal{A}} \hat{Q}_{t+1}^{i+k-t-1}(s_{t+1}^j, a))\}_{j=1}^m)$.
7:         **end for**
8:         Define $\pi_i$ by $\pi_i(s) \leftarrow \arg\max_a \hat{Q}_i^k(s, a)$.
9:     **end for**
10: **return** $\pi_1, \ldots, \pi_T$.
11: **end procedure**

---

a sequence of actions that solve a deterministic MDP by simulating a Monte Carlo planning algorithm. At each iteration, it collects $m$ episodes for each $k$-long action sequence by playing the previous learned actions, the $k$-long action sequence, and then sampling from $\pi^{\text{rand}}$. Then, it picks the action sequence with the highest mean return across the $m$ episodes and adds its first action to the sequence of learned actions.

At first, it seems very difficult to translate GORP to the stochastic setting. It learns an open-loop sequence of actions, while stochastic environments can only be solved by a closed-loop policy. It also relies on being able to repeatedly reach the same states to estimate their Q-values, which in a stochastic MDP is often impossible due to randomness in the transitions.

**Regressing the random policy's Q-function**     To understand how we overcome these challenges, start by considering the first iteration of GORP ($i = 1$) when $k = 1$. In this case, GORP simply estimates the Q-function of the random policy ($Q^1 = Q^{\pi^{\text{rand}}}$) at the fixed initial state $s_1$ for each action as an empirical average over random rollouts. The difficulty in stochastic environments is that the initial state $s_1$ is sampled from a distribution $p(s_1)$ instead of being fixed. How can we precisely estimate $Q^1(s_1, a)$ over a variety of states and actions when we may never sample the same initial state twice? Our key insight is to replace an *average* over random rollouts with *regression* of the Q-function from samples of the form $(s_1, a_1, y)$, where $y = \sum_{t=1}^T R_t(s_t, a_t)$. Standard regression algorithms attempt to estimate the conditional mean $E[y \mid s_1, a_1]$. Since in this case $E[y \mid s_1, a_1] = Q_1^1(s_1, a_1)$, if our regression algorithm works well then it should output $\hat{Q}_1^1 \approx Q_1^1$.

If we can precisely regress $\hat{Q}_1^1 \approx Q_1^1$, then for most states $s_1$ we should have $\arg\max_a \hat{Q}_1^1(s_1, a) \subseteq \arg\max_a Q_1^1(s_1, a)$. This, combined with the MDP being 1-QVI-solvable, means that by setting $\pi_1(s_1) \in \arg\max_a \hat{Q}_1^1(s_1, a)$, $\pi_1$ should take optimal actions most of the time. If we fix $\pi_1$ for the remainder of training, then this means there is a fixed distribution over $s_2$, meaning we can also regress $\hat{Q}_2^1 \approx Q_2^1$, and thus learn $\pi_2$, and so on for $\pi_3, \ldots, \pi_T$.

**Extending to $k - 1$ steps of Q iteration**     While this explains how to extend GORP to stochastic environments when $k = 1$, what about when $k > 1$? In this case, GORP follows the first action of the $k$-action sequence with the highest estimated return. However, in stochastic environments, it rarely makes sense to consider a fixed $k$-action sequence, since generally after taking one action the agent must choose its next action the specific state it reached. Thus, again it is unclear how to extend this part of GORP to the stochastic case. To overcome this challenge, we combine two insights. First, we can reformulate picking the (first action of the) action sequence with the highest estimated return as a series of Bellman backups, as shown in Figure 1a.

**Approximating backups with fitted Q iteration**     Our second insight is that we can implement these backups in stochastic environments via fitted-Q iteration (Ernst et al., 2005), which estimates $Q_t^j$ by regressing from samples of the form $(s_t, a_t, y)$, where $y = R_t(s_t, a_t) + \max_{a_{t+1} \in \mathcal{A}} Q_{t+1}^{j-1}(s_{t+1}, a_{t+1})$. Thus, we can implement the $k - 1$ backups of GORP by performing $k - 1$ steps of fitted-Q iteration. This allows us to extend GORP to stochastic environments when $k > 1$. Putting together these insights gives the shallow Q-iteration via reinforcement learning (SQIRL) algorithm, which is presented in full as Algorithm 2.

**Regression assumptions**     To implement the regression and FQI steps, SQIRL uses a *regression oracle* REGRESS($\{(s^j, a^j, y^j)_{j=1}^m\}$) which takes as input a dataset of tuples $(s^j, a^j, y^j)$ for $j \in [m]$

| Setting | Sample complexity bounds | |
| --- | --- | --- |
| | Strategic exploration | SQIRL |
| Tabular MDP | $\widetilde{O}(TSA/\epsilon^2)$ | $\widetilde{O}(kT^3SA^{\bar{H}_k+1}/\epsilon)$ |
| Linear MDP | $\widetilde{O}(T^2d^2/\epsilon^2)$ | $\widetilde{O}(kT^3dA^{\bar{H}_k}/\epsilon)$ |
| Q-functions with finite pseudo-dimension | — | $\widetilde{O}(k5^kT^3dA^{\bar{H}_k}/\epsilon)$ |

Table 1: A comparison of our bounds for the sample complexity of SQIRL with bounds from the literature on strategic exploration (Azar et al., 2017; Jin et al., 2018; 2021; Chen et al., 2022). SQIRL can solve stochastic MDPs with a sample complexity that is exponential only in the effective horizon $\bar{H}_k$. Since SQIRL only requires a regression oracle that can estimate Q-functions, it can be used with a broader range of function classes, including any with finite pseudo-dimension.

and outputs a function $\hat{Q} : \mathcal{S} \times \mathcal{A} \to [0,1]$ that aims to predict $E[y \mid s,a]$. In order to analyze the sample complexity of SQIRL, we require the regression oracle to satisfy some basic properties, which we formalize in the following assumption.

**Assumption 3.5** (Regression oracle conditions). *Suppose the codomain of the regression oracle* REGRESS$(\cdot)$ *is* $\mathcal{H}$. *Define* $\mathcal{V} = \{V(s) = \max_{a \in \mathcal{A}} Q(s,a) \mid Q \in \mathcal{H}\}$ *as the class of possible value functions induced by outputs of* REGRESS. *We assume there are functions* $F : (0,1] \to (0,\infty)$ *and* $G : [1,\infty) \times (0,1] \to (0,\infty)$ *such that the following conditions hold.*

**(Regression)** *Let* $Q = Q_t^1$ *for any* $t \in [T]$. *Suppose a dataset* $\{(s,a,y)\}_{j=1}^m$ *is sampled i.i.d. from a distribution* $\mathcal{D}$ *such that* $y \in [0,1]$ *almost surely and* $E_{\mathcal{D}}[y \mid s,a] = Q(s,a)$. *Then with probability greater than* $1 - \delta$ *over the sample,*

$$E_{\mathcal{D}}\big[(\hat{Q}(s,a) - Q(s,a))^2\big] \leq O(\tfrac{\log m}{m}F(\delta)) \qquad where \quad \hat{Q} = \text{REGRESS}\big(\{(s^j,a^j,y^j)\}_{j=1}^m\big).$$

**(Fitted Q-iteration)** *Let* $Q = Q_t^i$ *for any* $t \in [T-1]$ *and* $i \in [k-1]$; *define* $V(s) = \max_{a \in \mathcal{A}} Q_{t+1}^{i-1}(s,a)$. *Suppose a dataset* $\{(s,a,s')\}_{j=1}^m$ *is sampled i.i.d. from a distribution* $\mathcal{D}$ *such that* $s' \sim p_t(\cdot \mid s,a)$. *Then with probability greater than* $1 - \delta$ *over the sample, we have for all* $\hat{V} \in \mathcal{V}$ *uniformly,*

$$E_{\mathcal{D}}\big[(\hat{Q}(s,a) - Q(s,a))^2\big]^{1/2} \leq \alpha E_{\mathcal{D}}\big[(\hat{V}(s') - V(s'))^2\big]^{1/2} + O\Big(\sqrt{\tfrac{\log m}{m}G(\alpha,\delta)}\Big)$$

$$where \quad \hat{Q} = \text{REGRESS}(\{(s^j,a^j,R_t(s^j,a^j) + \hat{V}'(s'^j))\}_{j=1}^m).$$

While the conditions in Assumption 3.5 may seem complex, they are relatively mild. The first condition simply says that the regression oracle can take i.i.d. unbiased samples of the random policy's Q-function and accurately estimate it in-distribution. The error must decrease as $\widetilde{O}(F(\delta)/m)$ as the sample size $m$ increases for some $F(\delta)$ which depends on the regression oracle. The second condition is a bit more unusual. It controls how error propagates from an approximate value function at timestep $t+1$ to a Q-function estimated via FQI from the value function at timestep $t$. In particular, the assumption requires that the root mean square (RMS) error in the Q-function be at most $\alpha$ times the RMS error in the value function, plus an additional term of $\widetilde{O}(\sqrt{G(\alpha,\delta)/m})$ where $G(\alpha,\delta)$ can again depend on the regression oracle used.

In Appendix A, we show that a broad class of regression oracles satisfy Assumption 3.5, including least-squares regression in tabular MDPs, linear MDPs, and MDPs whose Q-functions are contained in a hypothesis class of finite pseudo-dimension. The latter case even includes MDPs whose Q-functions are representable by neural networks. For example, in linear MDPs, both conditions are satisfied with $\alpha = 1$ and $F(\delta) = G(1,\delta) = \widetilde{O}(d + \log(1/\delta))$.

Given a regression oracle that satisfies Assumption 3.5, we can prove our main theoretical result: the following upper bound on the sample complexity of SQIRL.

**Theorem 3.6** (SQIRL sample complexity). *Fix* $\alpha \geq 1$, $\delta \in (0,1]$, *and* $\epsilon \in (0,1]$. *Suppose* REGRESS *satisfies Assumption 3.5 and let* $D = F(\frac{\delta}{kT}) + G(\alpha, \frac{\delta}{kT})$. *Then if the MDP is* $k$-QVI-solvable for some $k \in [T]$, *there is a univeral constant* $C$ *such that SQIRL (Algorithm 2) will return an* $\epsilon$-optimal policy with probability at least $1 - \delta$ if $m \geq C\frac{kT\alpha^{2(k-1)}A^kD}{\Delta_k^2\epsilon}\log\frac{kT\alpha AD}{\Delta_k\epsilon}$. *Thus, the sample complexity of SQIRL is*

$$N_{\epsilon,\delta}^{SQIRL} = \widetilde{O}\left(kT^3\alpha^{2(k-1)}A^{\bar{H}_k}D\log(\alpha D)/\epsilon\right). \tag{1}$$

| Algorithm | Envs. solved |
|---|---|
| PPO | 96 |
| DQN | 76 |
| SQIRL | 69 |
| GORP | 26 |

Table 2: The number of sticky-action BRIDGE environments (out of 155) solved by four RL algorithms. Our SQIRL algorithm solves more than 2/3 of the environments that PPO does and nearly as many as DQN. Meanwhile, GORP (Laidlaw et al., 2023) fails in most because it is not designed for stochastic environments.

| Algorithms | | Sample complexity comparison | |
|---|---|---|---|
| | | Correl. | Median ratio |
| SQIRL | PPO | 0.83 | 1.38 |
| SQIRL | DQN | 0.56 | 1.00 |
| PPO | DQN | 0.48 | 0.59 |

Table 3: A comparison of the empirical sample complexities of SQIRL, PPO, and DQN in the sticky-action BRIDGE environments. SQIRL's sample complexity has higher Spearman correlation with PPO and DQN than they do with each other. Furthermore, SQIRL tends to have just slightly worse sample complexity then PPO and a bit better sample complexity than DQN.

To understand this bound on the sample complexity of SQIRL, compare it to GORP's sample complexity (Lemma 3.4). Like GORP, SQIRL has sample complexity exponential in only the effective horizon. As shown in Appendix A, in many cases we can set $\alpha = 1$ and $D \asymp d + \log(kT/\delta)$, where $d$ is the pseudo-dimension of the hypothesis class used by the regression oracle. Then, SQIRL's sample complexity of SQIRL is $\widetilde{O}(kT^3 A^{\bar{H}_k} d/\epsilon)$—ignoring log factors, just a $Td/\epsilon$ factor more than the sample complexity of GORP. The factor of $d$ is necessary because SQIRL must learn a Q-function that generalizes over many states, while GORP can estimate the Q-values at a single state in deterministic environments. The $1/\epsilon$ dependence on the desired suboptimality is standard for stochastic environments, e.g., see the strategic exploration bounds in Table 1. While our sample complexity bounds are exponential in $k$, in practice $k$ is quite small. Figure 2 shows many environments can be approximately solved with $k = 1$ and we run all experiments in Section 4 with $k \leq 5$.

## 4 EXPERIMENTS

While our theoretical results strongly suggest that SQIRL and the stochastic effective horizon can explain deep RL performance, we also want to validate these insights empirically. To do so, we implement SQIRL using deep neural networks for the regression oracle and compare its performance to two common deep RL algorithms, PPO (Schulman et al., 2017) and DQN (Mnih et al., 2015). We evaluate the algorithms in sticky-action versions of the BRIDGE environments from Laidlaw et al. (2023). These environments are a challenging benchmark for RL algorithms because they are stochastic and have high-dimensional states that necessitate neural network function approximation.

In practice, we slightly modify Algorithm 2 for use with deep neural networks. Following standard practice in deep RL, we use a single neural network to regress the Q-function across all timesteps, rather than using a separate Q-network for each timestep. However, we still "freeze" the greedy policy at each iteration (line 8 in Algorithm 2) by storing a copy of the network's weights from iteration $i$ and using it for acting on timestep $i$ in future iterations. Second, we stabilize training by using a replay buffer to store the data collected from the environment and then sampling batches from it to train the Q-network. Note that neither of these changes the core algorithm: our implementation is still entirely based around iteratively estimating $Q^k$ by using regression and fitted-Q iteration.

In each environment, we run PPO, DQN, SQIRL, and GORP for 5 million timesteps. We use the Stable-Baselines3 implementations of PPO and DQN (Raffin et al., 2021). During training, we evaluate the latest policy every 10,000 training timesteps for 100 episodes. We also calculate the exact optimal return of the sticky-action environments using the tabular representations from the BRIDGE dataset. If the mean evaluation return of the algorithm reaches the optimal return, we consider the algorithm to have solved the environment. We say the empirical sample complexity of the algorithm in the environment is the number of timesteps needed to reach that optimal return.

Since SQIRL and GORP take parameters $k$ and $m$, we need to tune these parameters for each environment. For each $k \in \{1, 2, 3, 4, 5\}$, we perform a binary search over values of $m$ to find the smallest value for which the algorithm solves the environment. We also slightly tune the hyperparameters of PPO and DQN; see Appendices C and D for all experiment details and results. We do not claim that SQIRL is as practical as PPO or DQN, since it requires much more hyperparameter tuning; instead, we mainly see SQIRL as a tool for understanding deep RL.

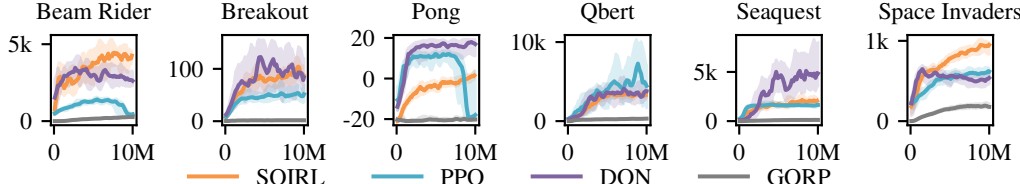

Figure 4: The performance of SQIRL in standard full-length Atari environments is comparable to PPO and DQN. This suggests that PPO and DQN succeed in standard benchmarks for similar reasons that SQIRL succeeds. Thus, our theoretical analysis of SQIRL based on the effective horizon can help explain deep RL performance in these environments.

The results of our experiments are shown in Tables 2 and 3 and Figure 3. Table 2 lists the number of environments solved by each algorithm. GORP barely solves any of the sticky-action BRIDGE environments, validating that our evaluation environments are stochastic enough that function approximation is necessary to solve them. In contrast, we find that SQIRL solves about two-thirds as many environments as PPO and nearly as many as DQN. This shows that SQIRL is not simply a useful algorithm in theory—it can solve a wide variety of stochastic environments in practice. It also suggests that the assumptions we introduce in Section 3 hold for RL in realistic environments with neural network function approximation. If the effective horizon was actually high, or if neural networks could not effectively regression the random policy's Q-function, we would not expect SQIRL to work as well as it does.

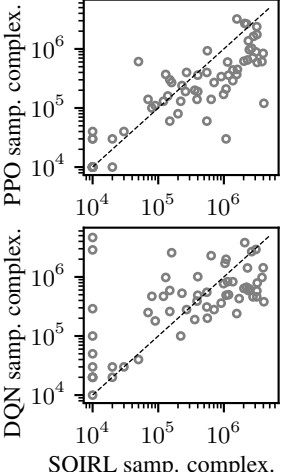

Figure 3: The empirical sample complexity of SQIRL correlates closely with that of PPO and DQN, suggesting that our theoretical analysis of SQIRL is a powerful tool for understanding when and why deep RL works in stochastic environments.

Table 2 and Figure 3 compare the empirical sample complexities of PPO, DQN, and SQIRL. In Table 2, we report the Spearman correlation between the sample complexities of each pair of algorithms in the environments they both solve. We find that SQIRL's sample complexity correlates better with that of PPO and DQN than they correlate with each other. We also report the median ratio of the sample complexities of each pair of algorithms to see if they agree in absolute scale. We find that SQIRL tends to have similar sample complexity to both PPO and DQN; it typically performs about the same as DQN and slightly worse than PPO. The fact that there is a close match between the performance of SQIRL and deep RL algorithms—when deep RL has low sample complexity, so does SQIRL, and vice versa—suggests that our theoretical explanation for why SQIRL succeeds is also a good explanation for why deep RL succeeds.

**Full-length Atari games** Besides the BRIDGE environments, which have relatively short horizons, we also compared SQIRL to deep RL algorithms in full-length Atari games. We use the standard Atari evaluation setup from Stable-Baselines3, except that we disable episode restart on loss of life as this does not fit into our RL formalism. A comparison of the learning curves for SQIRL, GORP, PPO, and DQN is shown in Figure 4. GORP performs poorly, but SQIRL performs comparably to PPO and DQN: it achieves higher reward than both PPO and DQN in three of the six games and worse reward than both in only two. This implies that our conclusions from the experiments in the BRIDGE environments are also applicable to more typical RL benchmark environments.

## 5 CONCLUSION

We have presented theoretical and empirical evidence that SQIRL and the effective horizon can help explain why deep RL succeeds in stochastic environments. Previous theoretical work has not satisfactorily explained why random exploration and complex function approximators should enable efficient RL. However, we leverage regression, fitted Q-iteration, and the low effective horizon assumption to close the theory-practice gap. We hope this paves the way for work that further advances our understanding of deep RL performance or builds improved algorithms based on our analysis.

ETHICS STATEMENT

The main goal of our research is to better understand theoretically when and why deep reinforcement learning succeeds and fails in practice. Since our work is not immediately useful for applications and is instead aimed at scientific understanding, we do not believe there are immediate ethics concerns.

REPRODUCIBILITY STATEMENT

We include details throughout the paper that can be used to reproduce our results. In particular, Section 4 and Appendix C contain the details of our empirical experiments. Section B contains the proofs of our theoretic results.

ACKNOWLEDGMENTS

We would like to thank Sam Toyer for feedback on drafts as well as Jiantao Jiao for helpful discussions.

This work was supported by a grant from Open Philanthropy to the Center for Human-Compatible Artificial Intelligence at UC Berkeley. Cassidy Laidlaw is supported by an Open Philanthropy AI Fellowship.

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

# APPENDIX

## A  LEAST-SQUARES REGRESSION ORACLES

In this appendix, we prove that many least-squares regression oracles satisfy Assumption 3.5 and thus can be used in SQIRL. These regression oracles minimize the empirical least-squares loss on the training data over some hypothesis class $\mathcal{H}$:

$$\text{REGRESS}(\{(s^j, a^j, y^j)\}_{j=1}^m) = \arg\min_{Q \in \mathcal{H}} \frac{1}{m} \sum_{j=1}^m \left(Q(s^j, a^j) - y^j\right)^2.$$

Proving that Assumption 3.5 is satisfied for such an oracle depends on some basic properties of $\mathcal{H}$. First, we require that $\mathcal{H}$ is of bounded complexity, since otherwise it is impossible to learn a Q-function that generalizes well. We formalize this by requiring a simple bound on the covering number of $\mathcal{H}$:

**Definition A.1.** *Suppose $\mathcal{H}$ is a hypothesis class of functions $Q : \mathcal{S} \times \mathcal{A} \to [0, 1]$. We say $\mathcal{H}$ is a* VC-type *hypothesis class if for any probability measure $P$ over $\mathcal{S} \times \mathcal{A}$, the $L_2(P)$ covering number of $\mathcal{H}$ is bounded as $N(\mathcal{H}, L_2(P); \epsilon) \le \left(\frac{B}{\epsilon}\right)^d$, where $\|Q - Q'\|_{L_2(P)}^2 = E_P[(Q(s, a) - Q'(s, a))^2]$.*

Many hypothesis classes are VC-type. For instance, if $\mathcal{H}$ has finite pseudo-dimension $d$, then it is VC-type with $d = d$ and $B = O(1)$. If $\mathcal{H}$ is parameterized by $\theta$ in a bounded subset of $\mathbb{R}^d$ and $Q^\theta$ is Lipschitz in its parameters, then $\mathcal{H}$ is also VC-type with $d = d$ and $B = O(\log(\rho L))$, where $\|\theta\|_2 \le \rho$ and $L$ is the Lipschitz constant. See Appendix A.1 for more information.

Besides bounding the complexity of $\mathcal{H}$, we also need it to be rich enough to fit the Q-functions in the MDP. We formalize this in the following two conditions.

**Definition A.2.** *We say $\mathcal{H}$ is $k$-realizable if for all $i \in [k]$ and $t \in [T]$, $Q_t^i \in \mathcal{H}$.*

**Definition A.3.** *We say $\mathcal{H}$ is closed under QVI if for any $t \in \{2, \dots, T\}$, $\hat{Q}_t \in \mathcal{H}$ implies that $QVI(\hat{Q}_t) \in \mathcal{H}$.*

Assuming that $\mathcal{H}$ is $k$-realizable is very mild: we would expect that function approximation-based RL would not work at all if the function approximators cannot fit Q-functions in the MDP. The second assumption, that $\mathcal{H}$ is closed under QVI, is more restrictive. However, it turns out this is not necessary for proving that Assumption 3.5 is satisfied; if $\mathcal{H}$ is not closed under QVI, then it just results in slightly worse sample complexity bounds.

**Theorem A.4.** *Suppose $\mathcal{H}$ is $k$-realizable and of VC-type for constants $B$ and $d$. Then least squares regression over $\mathcal{H}$ satisfies Assumption 3.5 with*

$$F(\delta) = O\left(d\log(Bd) + \log(1/\delta)\right)$$
$$G(\alpha, \delta) = O\left(\left(d\log(ABd/(\alpha - 2)) + \log(1/\delta)\right)/(\alpha - 2)^4\right).$$

*Furthermore, if $\mathcal{H}$ is also closed under QVI, then we can remove all $(\alpha - 2)$ factors in $G$.*

Theorem A.4 (proved in Appendix B.3) allows us to immediately bound the sample complexity bounds of SQIRL in a number of settings. For instance, consider a linear MDP with state-action features $\phi(s, a) \in \mathbb{R}^d$. We can let $\mathcal{H} = \{\hat{Q}(s, a) = w^\top \phi(s, a) \mid w^\top \phi(s, a) \in [0, 1] \quad \forall (s, a) \in \mathcal{S} \times \mathcal{A}\}$. This hypothesis class is realizable for any $k$, closed under QVI, and of VC-type, meaning SQIRL's sample complexity is at most $\widetilde{O}(kT^3 dA^{\bar{H}_k}/\epsilon)$. Since tabular MDPs are a special case of linear MDPs with $d = SA$, this gives bounds for the tabular case as well. Table 1 shows a comparison between these bounds and previously known bounds for *strategic* exploration.

However, our analysis can also handle much more general cases than any strategic exploration bounds in the literature. For instance, suppose $\mathcal{H}$ consists of neural networks with $n$ parameters and $\ell$ layers, and say that $\mathcal{H}$ is $k$-realizable, but not necessarily closed under QVI. Then $\mathcal{H}$ has pseudo-dimension of $d = O(n\ell \log(n))$ (Bartlett et al., 2017) and we can bound the sample complexity of SQIRL by $\widetilde{O}(k5^k T^3 n\ell A^{\bar{H}_k}/\epsilon)$, where we use $\alpha = \sqrt{5}$.

## A.1 VC-TYPE HYPOTHESIS CLASSES

We now describe two cases when hypothesis classes are of VC-type; thus, by Theorem A.4 these hypothesis classes satisfy Assumption 3.5 and can be used as part of SQIRL.

**Example A.5.** *We say $\mathcal{H}$ has pseudo-dimension $d$ if the collection of all subgraphs of the functions in $\mathcal{H}$ forms a class of sets with VC dimension $d$. Then by Theorem 2.6.7 of van der Vaart & Wellner (1996), $\mathcal{H}$ is a VC-type class with*

$$N(\mathcal{H}, L_2(P); \epsilon) \leq Cd(16e)^d \left(\frac{1}{\epsilon}\right)^{2(d-1)} = O\left(\left(\frac{4e}{\epsilon}\right)^{2d}\right).$$

**Example A.6.** *Suppose $\mathcal{H}$ is parameterized by $\theta \in \Theta \subseteq \mathbb{R}^d$ with $\|\theta\|_2 \leq B \; \forall \theta$, and $Q_\theta(s,a)$ is Lipschitz in $\theta$, i.e.,*

$$|Q_\theta(s,a) - Q_{\theta'}(s,a)| \leq L\|\theta - \theta'\|_2 \quad \forall \theta, \theta' \in \Theta.$$

*By Corollary 4.2.13 of Vershynin (2018), the $\epsilon$-covering number of $\{\theta \in \mathbb{R}^d \mid \|\theta\|_2 \leq B\}$ is bounded as $(1 + 2B/\epsilon)^d$. Therefore, the $\epsilon$-packing number of $\mathcal{H}$ is bounded as $(1 + 4B/\epsilon)^d$ (Lemma 4.2.8 of Vershynin (2018)); this in turn implies that the $\epsilon$ packing number of $\Theta$ is bounded identically, since any $\epsilon$-packing of $\Theta$ is also an $\epsilon$-packing of $\mathcal{H}$, which means that the $\epsilon$-covering number of $\Theta$ is also bounded as $(1 + 4B/\epsilon)^d$. If we take $\mathcal{N}_{\epsilon/L}$ to be an $\epsilon/L$-covering of $\Theta$, then for any $Q_\theta$, there must be some $\theta' \in \mathcal{N}_{\epsilon/L}$ such that $\|\theta - \theta'\|_2 \leq \epsilon/L$, which implies for any probability measure $P$ that*

$$\|Q_\theta(s,a) - Q_{\theta'}(s,a)\|_{L_2(P)} = E_P\left[(Q_\theta(s,a) - Q_{\theta'}(s,a))^2\right]^{1/2}$$
$$\leq E_P\left[L^2\|\theta - \theta'\|_2^2\right]^{1/2}$$
$$= L\|\theta - \theta'\|_2 \leq \epsilon.$$

*Thus $\{Q_\theta \mid \theta \in \mathcal{N}_{\epsilon/L}\}$ is an $\epsilon$-covering of $\mathcal{H}$, which implies that the $L_2(P)$ covering number of $\mathcal{H}$ is bounded as*

$$N(\mathcal{H}, L_2(P); \epsilon) \leq N(\Theta, L_2(P); \epsilon/L) \leq (1 + 4BL/\epsilon)^d = O\left(\left(\frac{4BL}{\epsilon}\right)^d\right).$$

## B PROOFS

### B.1 PROOF OF LEMMA 3.4

**Lemma 3.4.** *The deterministic effective horizon $H$ is bounded as*
$$H \leq \min_k \left[\bar{H}_k + \log_A O\left(\log\left(TA^k\right)\right)\right].$$

*Furthermore, if an MDP is $k$-QVI-solvable, then with probability at least $1 - \delta$, GORP will return an optimal policy with sample complexity at most $O(kT^2 A^{\bar{H}_k} \log(TA/\delta))$.*

*Proof.* The bound on $H$ follows immediately from Theorem 5.4 of Laidlaw et al. (2023) by noticing that in our setting, the Q and value functions are always upper-bounded by 1. The bound the sample complexity of GORP then follows from Lemma 5.3 of Laidlaw et al. (2023). ∎

### B.2 PROOF OF THEOREM 3.6

To prove our bounds on the sample complexity of SQIRL, we first introduce a series of auxiliary lemmas.

**Lemma B.1.** *Suppose that an MDP is $k$-QVI solvable and we iteratively find deterministic policies $\pi_1, \ldots, \pi_T$ such that for each $t$, $P_\pi(\pi_t(s_t) \notin \arg\max_a Q_t^k(s_t, a)) \leq \epsilon/T$, where states $s_t$ are sampled by following policies $\pi_1, \ldots, \pi_{t-1}$ for timesteps 1 to $t-1$. Then $\pi$ is $\epsilon$-optimal in the overall MDP, i.e.*
$$J(\pi) \geq \max_{\pi^*} J(\pi^*) - \epsilon.$$

*Proof.* Let $\mathcal{E}$ denote the event that there is some $t \in [T]$ when $a_t \notin \arg\max_a Q_t^k(s_t, a)$. By a union bound, we have $P_\pi(\mathcal{E}) \leq \epsilon$. Now, let $\pi^*$ be a policy in $\Pi(Q^k)$ that agrees with $\pi$ at all states and

timesteps where $\pi_t(s) \in \arg\max_a Q_t^k(s, a)$. We can write $\tilde{\mathcal{E}}$ as the event that $\exists t \in [T]$, $\pi(s_t) \neq \pi^*(s_t)$, which is equivalent to $\mathcal{E}$ under the distribution induced by $\pi$. We can now decompose $J(\pi)$ as

$$J(\pi) = E_\pi \left[ \sum_{t=1}^T R_t(s_t, a_t) \right]$$

$$\geq E_\pi \left[ \sum_{t=1}^T R_t(s_t, a_t) \mid \neg\mathcal{E} \right] P_\pi(\neg\mathcal{E})$$

$$= E_{\pi^*} \left[ \sum_{t=1}^T R_t(s_t, a_t) \mid \neg\tilde{\mathcal{E}} \right] P_\pi(\neg\mathcal{E})$$

$$= E_{\pi^*} \left[ \sum_{t=1}^T R_t(s_t, a_t) \mid \neg\tilde{\mathcal{E}} \right] P_\pi(\neg\mathcal{E}) + E_{\pi^*} \left[ \sum_{t=1}^T R_t(s_t, a_t) \mid \tilde{\mathcal{E}} \right] (P_\pi(\mathcal{E}) - P_\pi(\mathcal{E}))$$

$$= E_{\pi^*} \left[ \sum_{t=1}^T R_t(s_t, a_t) \right] - E_{\pi^*} \left[ \sum_{t=1}^T R_t(s_t, a_t) \mid \tilde{\mathcal{E}} \right] P_\pi(\mathcal{E})$$

$$\geq J(\pi^*) - \epsilon.$$

$\blacksquare$

**Lemma B.2.** *Let $\mathcal{D}$ be a distribution over states and actions such that $P_\mathcal{D}(a \mid s) = 1/A$ for all $s \in \mathcal{S}$, $a \in \mathcal{A}$. Then for any $Q$ and $\hat{Q} : \mathcal{S} \times \mathcal{A} \to [0, 1]$, defining $V(s) = \max_{a \in \mathcal{A}} Q(s, a)$ and $\hat{V}$ analogously, we have*

$$E_\mathcal{D} \left[ \left( \hat{V}(s) - V(s) \right)^2 \right] \leq A E_\mathcal{D} \left[ \left( \hat{Q}(s, a) - Q(s, a) \right)^2 \right].$$

*Proof.* We have

$$E_\mathcal{D} \left[ \left( \hat{V}(s) - V(s) \right)^2 \right] = E_\mathcal{D} \left[ \left( \max_{a \in \mathcal{A}} \hat{Q}(s, a) - \max_{a \in \mathcal{A}} Q(s, a) \right)^2 \right]$$

$$\leq E_\mathcal{D} \left[ \max_{a \in \mathcal{A}} \left( \hat{Q}(s, a) - Q(s, a) \right)^2 \right]$$

$$\leq E_\mathcal{D} \left[ \sum_{a \in \mathcal{A}} \left( \hat{Q}(s, a) - Q(s, a) \right)^2 \right]$$

$$= A E_\mathcal{D} \left[ \frac{1}{A} \sum_{a \in \mathcal{A}} \left( \hat{Q}(s, a) - Q(s, a) \right)^2 \right]$$

$$= A E_\mathcal{D} \left[ \left( \hat{Q}(s, a) - Q(s, a) \right)^2 \right],$$

where the final equality follows from the fact that $P_\mathcal{D}(a \mid s) = 1/A$. $\blacksquare$

**Lemma B.3.** *Suppose the MDP is $k$-QVI solvable and let $\pi_i$ be the policy constructed by stochastic GORP at timestep $i$. Then with probability at least $1 - \delta/T$,*

$$P_\pi \left( \pi_i(s) \notin \arg\max_a Q_i^k(s, a) \right) \leq O \left( \frac{\alpha^{2k-2} A^k \left( F(\frac{\delta}{kT}) + G(\alpha, \frac{\delta}{kT}) \right) \log m}{m \Delta_k^2} \right).$$

*Proof.* Let all expectations and probabilities $E$ and $P$ be with respect to the distribution of states and actions induced by following $\pi_1, \ldots, \pi_{i-1}$ for $t < i$ and $\pi^{\text{rand}}$ thereafter. To simplify notation, we write for any $Q_t, Q_t' : \mathcal{S} \times \mathcal{A} \to [0, 1]$ or $V_t, V_t' : \mathcal{S} \to [0, 1]$,

$$\|Q_t - Q_t'\|_2^2 = E \left[ (Q_t(s_t, a_t) - Q_t'(s_t, a_t))^2 \right]$$

$$\|V_t - V_t'\|_2^2 = E \left[ (V_t(s_t) - V_t'(s_t))^2 \right].$$

Let $\hat{V}_t^{i+k-t}(s) = \max_{a \in \mathcal{A}} \hat{Q}_t^{i+k-t}(s, a)$ and $V_t^{i+k-t}(s) = \max_{a \in \mathcal{A}} Q_t^{i+k-t}(s, a)$. Consider the following three facts:

1. By Assumption 3.5 part 1, with probability at least $1 - \delta/(kT)$,

$$\left\| \hat{Q}_{i+k-1}^1 - Q_{i+k-1}^1 \right\|_2^2 \leq C_1 \frac{F(\frac{\delta}{kT}) \log m}{m}.$$

2. By Lemma B.2, for all $t \in \{2, \ldots, k\}$,

$$\left\| \hat{V}_t^{i+k-t} - V_t^{i+k-t} \right\|_2^2 \leq A \left\| \hat{Q}_t^{i+k-t} - Q_t^{i+k-t} \right\|_2^2.$$

3. By Assumption 3.5 part 2, for any $t \in \{1, \ldots, k-1\}$, with probability at least $1 - \delta/(kT)$

$$\left\| \hat{Q}_t^{i+k-t} - Q_t^{i+k-t} \right\|_2 \leq \alpha \left\| \hat{V}_{t+1}^{i+k-t-1} - V_{t+1}^{i+k-t-1} \right\|_2 + \sqrt{C_2 \frac{G(\alpha, \frac{\delta}{kT}) \log m}{m}}.$$

   Note that it is key that this bound is uniform over all $\hat{V}_{t+1}^{i+k-t-1} \in \mathcal{V}$, since $\hat{V}_{t+1}^{i+k-t-1}$ is estimated based on the same data used to regress $\hat{Q}_t^{i+k-t}$.

Via a union bound all of the above facts hold with probability at least $1 - \delta/T$. We will combine them to recursively show for $t \in \{1, \ldots, k\}$,

$$\left\| \hat{Q}_t^{i+k-t} - Q_t^{i+k-t} \right\|_2 \leq \left( 4(\alpha\sqrt{A})^{k-t} - 3 \right) \sqrt{\frac{D \log m}{m}} \qquad \text{where} \quad D = C_1 F(\tfrac{\delta}{kT}) + C_2 G(\alpha, \tfrac{\delta}{kT}). \tag{2}$$

The base case $t = k$ is true by fact 1. Now let $t < k$ and assume the above holds for $t + 1$. By facts 2 and 3,

$$\begin{aligned}
\left\| \hat{Q}_t^{i+k-t} - Q_t^{i+k-t} \right\|_2 &\leq \alpha \left\| \hat{V}_{t+1}^{i+k-t-1} - V_{t+1}^{i+k-t-1} \right\|_2 + \sqrt{C_2 \frac{G(\alpha, \frac{\delta}{kT}) \log m}{m}} \\
&\leq \alpha\sqrt{A} \left\| \hat{Q}_{t+1}^{i+k-t-1} - Q_{t+1}^{i+k-t-1} \right\|_2 + \sqrt{C_2 \frac{G(\alpha, \frac{\delta}{kT}) \log m}{m}} \\
&\leq \alpha\sqrt{A} \left( 4(\alpha\sqrt{A})^{k-t-1} - 3 \right) \sqrt{\frac{D \log m}{m}} + \sqrt{\frac{D \log m}{m}} \\
&= \left( 4(\alpha\sqrt{A})^{k-t} - 3\alpha\sqrt{A} + 1 \right) \sqrt{\frac{D \log m}{m}} \\
&\leq \left( 4(\alpha\sqrt{A})^{k-t} - 3 \right) \sqrt{\frac{D \log m}{m}},
\end{aligned}$$

where the last inequality follows from $A \geq 2$ and $\alpha \geq 1$. Thus, by setting $t = i$ in (2), we see that with probability at least $1 - \delta/T$,

$$\left\| \hat{Q}_i^k - Q_i^k \right\|_2^2 \leq O\left( \frac{\alpha^{2k-2} A^{k-1} \left( F(\frac{\delta}{kT}) + G(\alpha, \frac{\delta}{kT}) \right) \log m}{m} \right) \tag{3}$$

$$P_\pi(\pi_i(s_i) \notin \arg\max_a Q_i^k(s_i, a)$$

$$\leq P_\pi \left( \arg\max_a \hat{Q}_i^k(s_i, a) \nsubseteq \arg\max_a Q_i^k(s_i, a) \right)$$

$$\overset{(i)}{\leq} P_\pi \left( \exists a \in \mathcal{A} \quad \text{s.t.} \quad \left| \hat{Q}_i^k(s_i, a) - Q_i^k(s_i, a) \right| \geq \Delta_k/2 \right)$$

$$\leq \sum_{a \in \mathcal{A}} P_\pi \left( \left| \hat{Q}_i^k(s_i, a) - Q_i^k(s_i, a) \right| \geq \Delta_k/2 \right)$$

$$= A \left( \frac{1}{A} \sum_{a \in \mathcal{A}} P_\pi \left( \left| \hat{Q}_i^k(s_i, a) - Q_i^k(s_i, a) \right| \geq \Delta_k/2 \right) \right)$$

$$= A P_\pi \left( \left| \hat{Q}_i^k(s_i, a_i) - Q_i^k(s_i, a_i) \right| \geq \Delta_k/2 \right)$$

$$\overset{(ii)}{\leq} \frac{A}{(\Delta_k/2)^2} E_\pi \left[ \left( \hat{Q}_i^k(s_i, a_i) - Q_i^k(s_i, a_i) \right)^2 \right]$$

$$= \frac{A}{(\Delta_k/2)^2} \left\| \hat{Q}_i^k - Q_i^k \right\|_2^2$$

$$= O \left( \frac{\alpha^{2k-2} A^k \left( F(\frac{\delta}{kT}) + G(\alpha, \frac{\delta}{kT}) \right) \log m}{m \Delta_k^2} \right).$$

Here, (i) follows from Definition 3.2 of the $k$-gap and (ii) follows from Markov's inequality. ∎

**Theorem 3.6** (SQIRL sample complexity). *Fix $\alpha \geq 1$, $\delta \in (0, 1]$, and $\epsilon \in (0, 1]$. Suppose REGRESS satisfies Assumption 3.5 and let $D = F(\frac{\delta}{kT}) + G(\alpha, \frac{\delta}{kT})$. Then if the MDP is $k$-QVI-solvable for some $k \in [T]$, there is a univeral constant $C$ such that SQIRL (Algorithm 2) will return an $\epsilon$-optimal policy with probability at least $1 - \delta$ if $m \geq C \frac{kT\alpha^{2(k-1)} A^k D}{\Delta_k^2 \epsilon} \log \frac{kT\alpha AD}{\Delta_k \epsilon}$. Thus, the sample complexity of SQIRL is*

$$N_{\epsilon,\delta}^{SQIRL} = \widetilde{O} \left( kT^3 \alpha^{2(k-1)} A^{\bar{H}_k} D \log(\alpha D)/\epsilon \right). \tag{1}$$

*Proof.* Given the lower bound on $m$, we can bound

$$\frac{\log m}{m} = O \left( \frac{\Delta_k^2 \epsilon}{T\alpha^{2k-2} A^k D} \right).$$

Combining this with Lemma B.3, we see that with probability at least $1 - \delta$, for all $i \in [T]$

$$P_\pi \left( \pi_i(s_i) \notin \arg\max_a Q_i^k(s_i, a) \right) \leq \epsilon/T.$$

Thus, by Lemma B.1, $\pi$ is $\epsilon$-optimal in the overall MDP $\mathcal{M}$. ∎

## B.3 PROOF OF THEOREM A.4

**Theorem A.4.** *Suppose $\mathcal{H}$ is $k$-realizable and of VC-type for constants $B$ and $d$. Then least squares regression over $\mathcal{H}$ satisfies Assumption 3.5 with*

$$F(\delta) = O \left( d \log(Bd) + \log(1/\delta) \right)$$
$$G(\alpha, \delta) = O \left( \left( d \log(ABd/(\alpha - 2)) + \log(1/\delta) \right) / (\alpha - 2)^4 \right).$$

*Furthermore, if $\mathcal{H}$ is also closed under QVI, then we can remove all $(\alpha - 2)$ factors in $G$.*

*Proof.* Throughout the proof, we will use the notation that $\|Q - Q'\|_2^2 = E_\mathcal{D}[(Q(s, a) - Q'(s, a))^2]$ and $\|V - V'\|_2^2 = E_\mathcal{D}[(V(s') - V'(s'))^2]$.

First, we will prove the regression part of Assumption 3.5. To do so, we use results on least-squares regression from Koltchinskii (2006). Note that our definition of VC-type classes coincides with condition (2.1) in Koltchinskii (2006). By combining Example 3 from Section 2.5 and Theorem

13 of Koltchinskii (2006), we have that for any $\bar{Q}(s,a)$ with $\mathbb{E}[y \mid s,a] = \bar{Q}(s,a)$, and for any $\lambda \in (0,1]$,

$$\mathbb{P}\left(\left\|\hat{Q} - \bar{Q}\right\|_2^2 \leq (1+\lambda) \inf_{\tilde{Q} \in \mathcal{H}} \left\|\tilde{Q} - \bar{Q}\right\|_2^2 + O\left(\frac{d}{m\lambda^2}\log\left(\frac{Bdm}{\lambda}\right) + \frac{u+1}{\lambda m}\right)\right) \leq \log\left(\frac{em}{u}\right)e^{-u}.$$

where $\qquad \hat{Q} = \text{REGRESS}(\{(s^j, a^j, y^j)\}_{j=1}^m). \qquad\qquad (4)$

If we set

$$u = \log\left(\frac{e + \log m}{\delta}\right) \geq 1,$$

then the right-hand side of (4) is bounded as

$$\log\left(\frac{em}{u}\right)e^{-u} \leq \log(em)e^{-u} = \log(em)\frac{\delta}{e + \log m} < \delta.$$

Thus, plugging this value of $u$ into (4), we have that with probability at least $1 - \delta$,

$$\left\|\hat{Q} - \bar{Q}\right\|_2^2 \leq (1+\lambda) \inf_{\tilde{Q} \in \mathcal{H}} \left\|\tilde{Q} - \bar{Q}\right\|_2^2 + O\left(\frac{d\log\left(\frac{Bdm}{\lambda}\right) + \log\frac{m}{\delta}}{m\lambda^2}\right). \qquad (5)$$

For the regression condition of Assumption 3.5, we have $\bar{Q} = Q_t^k \in \mathcal{H}$. Thus, $\inf_{\tilde{Q} \in \mathcal{H}} \|\tilde{Q} - \bar{Q}\|_2^2 = 0$, and we can set $\lambda = 1$ in (5) to obtain

$$\left\|\hat{Q} - Q_t^k\right\|_2^2 \leq O\left(\frac{d\log(Bdm) + \log\frac{m}{\delta}}{m}\right),$$

leading to the desired bound of

$$F(\delta) = O\left(d\log(Bd) + \log\frac{1}{\delta}\right).$$

To the fitted Q-iteration condition of Assumption 3.5, we begin by defining a norm $\rho$ on $\mathcal{V} \times \mathcal{H}$ by

$$\rho\big((V,Q),(V',Q')\big) = \max\{\|V - V'\|_2, \|Q - Q'\|_2\}.$$

Note that since we showed in Lemma B.2 that

$$E_\mathcal{D}\left[(V(s') - V'(s'))^2\right] \leq A E_{\mathcal{D}, a' \sim \text{Unif}(\mathcal{A})}\left[(Q(s',a') - Q'(s',a'))^2\right]$$
$$\text{where} \quad V(s') = \max_{a' \in \mathcal{A}} Q(s',a'), V'(s') = \max_{a' \in \mathcal{A}} Q'(s',a'),$$

this implies that any $\epsilon$-cover of $\mathcal{H}$ is also an $\epsilon\sqrt{A}$-cover of $\mathcal{V}$ with respect to $L_2(P)$ for any distribution $P$ over $s'$. Thus, by the definition of VC-type classes, we have

$$N(\mathcal{V}, L_2(\mathcal{D}); \epsilon) \leq \left(\frac{B\sqrt{A}}{\epsilon}\right)^d$$

$$N(\mathcal{H}, L_2(\mathcal{D}); \epsilon) \leq \left(\frac{B}{\epsilon}\right)^d$$

$$N(\mathcal{V} \times \mathcal{H}, \rho; \epsilon) \leq \left(\frac{B\sqrt{A}}{\epsilon}\right)^d \left(\frac{B}{\epsilon}\right)^d \leq \left(\frac{B\sqrt{A}}{\epsilon}\right)^{2d}.$$

Now define $\mathcal{W} \subseteq \mathcal{V} \times \mathcal{H}$ as

$$\mathcal{W} = \left\{(\hat{V}, \hat{Q}) \in \mathcal{V} \times \mathcal{H} \,\middle|\, \hat{Q} = \text{REGRESS}\left(\left\{(s^j, a^j, R_t(s^j, a^j) + \hat{V}(s'^j))\right\}_{j=1}^m\right)\right\}.$$

By properties of packing and covering numbers, since any $\epsilon$-packing of $\mathcal{W}$ is also an $\epsilon$-packing of $\mathcal{V} \times \mathcal{H}$, we have

$$N(\mathcal{W}, \rho; \epsilon) \leq M(\mathcal{W}, \rho; \epsilon/2) \leq M(\mathcal{V} \times \mathcal{H}, \rho; \epsilon/2) \leq N(\mathcal{V} \times \mathcal{H}, \rho; \epsilon/2) \leq \left(\frac{2B\sqrt{A}}{\epsilon}\right)^{2d}.$$

Thus, let $\mathcal{N}_{1/\sqrt{m}}$ be a $1/\sqrt{m}$-covering of $\mathcal{W}$ with size at most $(2B\sqrt{Am})^{2d}$.

Fix any $(\hat{V}, \hat{Q}) \in \mathcal{N}_{1/\sqrt{m}}$ and define $\bar{Q}(s,a) = E[R_t(s,a) + \hat{V}(s') \mid s,a]$. Then by an identical argument to (5), with probability at least $1 - \delta$, for any $\lambda \in (0,1]$,

$$\left\| \hat{Q} - \bar{Q} \right\|_2^2 \le (1 + \lambda) \inf_{\tilde{Q} \in \mathcal{H}} \left\| \tilde{Q} - \bar{Q} \right\|_2^2 + O\left( \frac{d \log\left( \frac{Bdm}{\lambda} \right) + \log \frac{m}{\delta}}{m\lambda^2} \right).$$

We can extend this to a bound on all $(\hat{V}, \hat{Q}) \in \mathcal{N}_{1/\sqrt{m}}$ by dividing $\delta$ by $|\mathcal{N}_{1/\sqrt{m}}|$ and applying a union bound. Thus, with probability at least $1 - \delta$, for all $(\hat{V}, \hat{Q}) \in \mathcal{N}_{1/\sqrt{m}}$ and any $\lambda \in (0,1]$,

$$\left\| \hat{Q} - \bar{Q} \right\|_2^2 \le (1 + \lambda) \inf_{\tilde{Q} \in \mathcal{H}} \left\| \tilde{Q} - \bar{Q} \right\|_2^2 + O\left( \frac{d \log\left( \frac{Bdm}{\lambda} \right) + d \log(BAm) + \log \frac{m}{\delta}}{m\lambda^2} \right)$$

$$= (1 + \lambda) \inf_{\tilde{Q} \in \mathcal{H}} \left\| \tilde{Q} - \bar{Q} \right\|_2^2 + O\left( \frac{d \log\left( \frac{BAdm}{\lambda} \right) + \log \frac{m}{\delta}}{m\lambda^2} \right).$$

Finally, we extend this to a bound over all $(\hat{V}, \hat{Q}) \in \mathcal{W}$. For any $(\hat{V}, \hat{Q}) \in \mathcal{W}$, there must be some $(\hat{V}', \hat{Q}') \in \mathcal{N}_{1/\sqrt{m}}$ such that $\rho((\hat{V}, \hat{Q}), (\hat{V}', \hat{Q}')) \le 1/\sqrt{m}$. Let $\bar{Q}'(s,a) = E[R_t(s,a) + \hat{V}'(s') \mid s,a]$. Then

$$\|\bar{Q} - \bar{Q}'\|_2^2 = E_{\mathcal{D}}\left[ \left( \bar{Q}(s,a) - \bar{Q}'(s,a) \right)^2 \right]$$

$$= E_{\mathcal{D}}\left[ \left( E_{\mathcal{D}}\left[ \hat{V}(s') - \hat{V}'(s') \right] \right)^2 \right]$$

$$\le E_{\mathcal{D}}\left[ \left( \hat{V}(s') - \hat{V}'(s') \right)^2 \right] \le \frac{1}{m},$$

where the second-to-last inequality follows from Jensen's inequality. Thus, by the triangle inequality,

$$\left\| \hat{Q} - \bar{Q} \right\|_2 \le \left\| \hat{Q} - \hat{Q}' \right\|_2 + \left\| \hat{Q}' - \bar{Q}' \right\|_2 + \left\| \bar{Q}' - \bar{Q} \right\|_2$$

$$\le \left\| \hat{Q}' - \bar{Q}' \right\|_2 + \frac{2}{\sqrt{m}}$$

$$\le \sqrt{1 + \lambda} \inf_{\tilde{Q}' \in \mathcal{H}} \left\| \tilde{Q}' - \bar{Q}' \right\|_2 + O\left( \sqrt{\frac{d \log\left( \frac{BAdm}{\lambda} \right) + \log \frac{m}{\delta}}{m\lambda^2}} \right).$$

for all $(\hat{V}, \hat{Q}) \in \mathcal{W}$ and any $\lambda \in (0,1]$ with probability at least $1 - \delta$.

We now consider the two possible conditions in the theorem. If $\mathcal{H}$ is both $k$-realizable and closed under QVI, then this implies $\bar{Q}' \in \mathcal{H}$ for all $(\hat{V}', \hat{Q}') \in \mathcal{N}_{1/\sqrt{m}}$, meaning $\inf_{\tilde{Q}' \in \mathcal{H}} \|\tilde{Q}' - \bar{Q}'\|_2 = 0$. Thus, we can set $\lambda = 1$ in the above bound to obtain

$$\left\| \hat{Q} - Q \right\|_2 \le \left\| \bar{Q} - Q \right\|_2 + \left\| \hat{Q} - \bar{Q} \right\|_2$$

$$\le \left\| \hat{V} - V \right\|_2 + O\left( \sqrt{\frac{d \log\left( \frac{BAdm}{\lambda} \right) + \log \frac{m}{\delta}}{m\lambda^2}} \right),$$

showing that the FQI condition of Assumption 3.5 holds with

$$G(\alpha, \delta) = O\left( d \log(BAd) + \log(1/\delta) \right).$$

Otherwise, if $\mathcal{H}$ is only $k$-realizable, then this implies $Q \in \mathcal{H}$. Thus,

$$\inf_{\tilde{Q}' \in \mathcal{H}} \left\| \tilde{Q}' - \bar{Q}' \right\|_2 \le \|Q - \bar{Q}'\|_2 \le \left\| V - \hat{V} \right\|_2 + \frac{1}{\sqrt{m}}.$$

This implies that

$$
\begin{aligned}
\left\|\hat{Q} - Q\right\|_2 &\leq \left\|\bar{Q} - Q\right\|_2 + \left\|\hat{Q} - \bar{Q}\right\|_2 \\
&\leq \left\|\hat{V} - V\right\|_2 + \sqrt{1+\lambda}\left\|V - \hat{V}\right\|_2 + \frac{1}{\sqrt{m}} + O\left(\sqrt{\frac{d\log\left(\frac{BAdm}{\lambda}\right) + \log\frac{m}{\delta}}{m\lambda^2}}\right) \\
&\leq (2 + \sqrt{\lambda})\left\|\hat{V} - V\right\|_2 + O\left(\sqrt{\frac{d\log\left(\frac{BAdm}{\lambda}\right) + \log\frac{m}{\delta}}{m\lambda^2}}\right).
\end{aligned}
$$

Setting $\alpha = 2 + \sqrt{\lambda}$ shows that the FQI condition of Assumption 3.5 holds with

$$
G(\alpha, \delta) = O\left(\frac{d\log\left(\frac{BAd}{\alpha - 2}\right) + \log\frac{1}{\delta}}{(\alpha - 2)^4}\right).
$$

∎

## C  EXPERIMENT DETAILS

In this appendix, we describe details of the experiments from Section 4. We use the implementations of PPO and DQN from Stable-Baselines3 (Raffin et al., 2021), and in general use their hyperparameters which have been optimized for Atari games. For network archictures, we use convolutional neural nets similar to those used by Mnih et al. (2015). We use a discount rate of $\gamma = 1$ for the Atari and Procgen environments in BRIDGE but $\gamma = 0.99$ for the MiniGrid environments, as otherwise we found that RL completely failed. We run 5 random seeds of each RL algorithm for each hyperparameter setting, recording the median reward and sample complexity.

**PPO**  We use the following hyperparameters for PPO:

| Hyperparameter | Value |
|---|---|
| Training timesteps | 5,000,000 |
| Rollout length | $\{128, 1, 280\}$ |
| SGD minibatch size | 256 |
| SGD epochs per iteration | 4 |
| Optimizer | Adam |
| Learning rate | $2.5 \times 10^{-4}$ |
| GAE coefficient ($\lambda$) | 0.95 |
| Entropy coefficient | 0.01 |
| Clipping parameter | 0.1 |
| Value function coefficient | 0.5 |

Table 4: Hyperparameters we use for PPO.

For each environment, we take the rollout length from $\{128, 1280\}$, as we found this was the most important parameter to tune.

**DQN**  We use the following hyperparameters for DQN:

| Hyperparameter | Value |
|---|---|
| Training timesteps | 5,000,000 |
| Timesteps before learning starts | 0 |
| Replay buffer size | 100,000 |
| Target network update frequency | 8,000 |
| Final $\epsilon$ | 0.01 |
| SGD minibatch size | 32 |
| Env. steps per gradient step | 4 |
| Optimizer | Adam |
| Learning rate | $10^{-4}$ |

Table 5: Hyperparameters we use for DQN.

We try decaying the $\epsilon$ value for $\epsilon$-greedy over the course of either 500 thousand or 5 million timesteps, as we found this was the most sensitive hyperparameter to tune for DQN.

**SQIRL**     We use the following hyperparameters for SQIRL:

| Hyperparameter | Value |
|---|---|
| Training timesteps | 5,000,000 |
| Replay buffer size | 1,000,000 |
| $k$ | $\{1, 2, 3, 4, 5\}$ |
| SGD minibatch size | 128 |
| SGD epochs per iteration | 10 |
| Optimizer | Adam |
| Learning rate | $10^{-4}$ |
| Loss weighting exponential smoothing | 0.99 |

Table 6: Hyperparameters we use for SQIRL.

As we describe in the main text, we run SQIRL with $k \in \{1, 2, 3, 4, 5\}$ and tune $m$ via binary search. As also described in the main text, we slightly modify SQIRL from Algorithm 2 for use with neural networks. We use a single Q-network with $k \cdot A$ outputs to represent $Q^1, \ldots, Q^k$. At each iteration, after collecting $m$ episodes according to Algorithm 2, we store tuples of the form $(s, a, s', r, y)$ in a replay buffer, where $s$ and $a$ are the action taken at some timestep, $s'$ is the next observed state (or $\perp$ if the end of the episode was reached), $r$ is the reward received, and $y$ is the observed reward-to-go summed over the remainder of the episode. The replay buffer keeps one million of the most recently observed transitions. Then, we sample $nmT$ transitions from the replay buffer in minibatches, where $n$ is the number of epochs (10 in our experiments).

For each minibatch we take a gradient step on the mean squared errors for $Q^1, \ldots, Q^k$. We find that the loss magnitudes can vary greatly between $Q^j$ for various $j$, since $Q^1$ has much higher-variance targets (the Monte Carlo reward-to-go) while $Q^2, Q^3, \ldots$ have lower-variance targets (the bootstrapped Q-value estimates). Thus, we divide each loss by an exponentially weighted average of its past values so that they all have roughly equal magnitude before averaging them and taking a gradient step with Adam.

After completing $n$ epochs of optimization for iteration $i$, we freeze the current Q-network weights and store them so that we can recall the greedy policy $\pi_i(s) = \arg\max_a \hat{Q}_i^k(s, a)$.

**GORP**     Similarly to SQIRL, to tune GORP on the sticky-action BRIDGE environments we consider $k \in \{1, 2, 3, 4, 5\}$ and determine the optimal $m$ for each via binary search.

**Full-horizon Atari games**     For the full-horizon Atari experiments, we run 5 random seeds and plot the median and range of *evaluation* returns—that is, returns from running the current greedy policy for 20 episodes every 100,000 training steps. We use the standard Stable-Baselines3 Atari environments with sticky actions except that we do not truncate episodes on loss-of-life. Truncating like this causes the initial state distribution for the next episode to be dependent on the policy used for the previous episode, which is nonstandard in RL and does not fit into our formalism. We train for 10 million steps and use a discount of $\gamma = 0.99$. While for the sticky-action BRIDGE environments we

Stable-Baseline3's `CnnPolicy`, for full-horizon Atari games we use the Impala CNN architecture (Espeholt et al., 2018).

For PPO and DQN, we use the same hyperparameters as above except for the following changes: for PPO, we use rollout length 128 and for DQN, we decay $\epsilon$ over the first one million timesteps. For SQIRL, we again tune $k \in \{1, 2, 3, 4, 5\}$. For SQIRL, we use $m = 10$ for all the games, except for Qbert, where we use $m = 20$. For GORP, we tune $k \in \{1, 2, 3\}$ and $m \in \{1, 3, 10\}$. We find that the following parameters are optimal for each game:

| Game | Optimal parameters for GORP | |
|---|---|---|
| | $k$ | $m$ |
| Beam Rider | 1 | 1 |
| Breakout | 3 | 1 |
| Pong | 1 | 3 |
| Qbert | 1 | 10 |
| Seaquest | 1 | 1 |
| Space Invaders | 1 | 3 |

# D  FULL RESULTS

## D.1  TABLE OF EMPIRICAL SAMPLE COMPLEXITIES

This table lists the empirical sample complexities of PPO, DQN, SQIRL, and GORP.

| | PPO | DQN | SQIRL | GORP |
|---|---|---|---|---|
| ALIEN$_{10}$ | $> 5 \times 10^6$ | $> 5 \times 10^6$ | $> 5 \times 10^6$ | $> 5 \times 10^6$ |
| AMIDAR$_{20}$ | $> 5 \times 10^6$ | $> 5 \times 10^6$ | $> 5 \times 10^6$ | $> 5 \times 10^6$ |
| ASSAULT$_{10}$ | $1.00 \times 10^4$ | $2.90 \times 10^5$ | $1.00 \times 10^4$ | $4.00 \times 10^4$ |
| ASTERIX$_{10}$ | $2.00 \times 10^5$ | $1.90 \times 10^5$ | $3.60 \times 10^5$ | $> 5 \times 10^6$ |
| ASTEROIDS$_{10}$ | $> 5 \times 10^6$ | $> 5 \times 10^6$ | $> 5 \times 10^6$ | $> 5 \times 10^6$ |
| ATLANTIS$_{10}$ | $3.00 \times 10^4$ | $1.00 \times 10^5$ | $1.00 \times 10^4$ | $1.40 \times 10^5$ |
| ATLANTIS$_{20}$ | $4.50 \times 10^5$ | $> 5 \times 10^6$ | $1.61 \times 10^6$ | $> 5 \times 10^6$ |
| ATLANTIS$_{30}$ | $> 5 \times 10^6$ | $> 5 \times 10^6$ | $> 5 \times 10^6$ | $> 5 \times 10^6$ |
| ATLANTIS$_{40}$ | $> 5 \times 10^6$ | $> 5 \times 10^6$ | $> 5 \times 10^6$ | $> 5 \times 10^6$ |
| ATLANTIS$_{50}$ | $> 5 \times 10^6$ | $> 5 \times 10^6$ | $> 5 \times 10^6$ | $> 5 \times 10^6$ |
| ATLANTIS$_{70}$ | $> 5 \times 10^6$ | $> 5 \times 10^6$ | $> 5 \times 10^6$ | $> 5 \times 10^6$ |
| BANKHEIST$_{10}$ | $8.90 \times 10^5$ | $2.92 \times 10^6$ | $3.08 \times 10^6$ | $> 5 \times 10^6$ |
| BATTLEZONE$_{10}$ | $1.40 \times 10^5$ | $2.29 \times 10^6$ | $6.40 \times 10^5$ | $> 5 \times 10^6$ |
| BEAMRIDER$_{20}$ | $> 5 \times 10^6$ | $4.34 \times 10^6$ | $> 5 \times 10^6$ | $> 5 \times 10^6$ |
| BOWLING$_{30}$ | $6.10 \times 10^5$ | $> 5 \times 10^6$ | $> 5 \times 10^6$ | $> 5 \times 10^6$ |
| BREAKOUT$_{10}$ | $1.40 \times 10^5$ | $2.50 \times 10^5$ | $7.00 \times 10^4$ | $1.00 \times 10^4$ |
| BREAKOUT$_{20}$ | $1.90 \times 10^5$ | $1.01 \times 10^6$ | $4.00 \times 10^5$ | $1.56 \times 10^6$ |
| BREAKOUT$_{30}$ | $1.63 \times 10^6$ | $> 5 \times 10^6$ | $2.83 \times 10^6$ | $> 5 \times 10^6$ |
| BREAKOUT$_{40}$ | $2.27 \times 10^6$ | $> 5 \times 10^6$ | $> 5 \times 10^6$ | $> 5 \times 10^6$ |
| BREAKOUT$_{50}$ | $2.06 \times 10^6$ | $> 5 \times 10^6$ | $> 5 \times 10^6$ | $> 5 \times 10^6$ |
| BREAKOUT$_{70}$ | $2.52 \times 10^6$ | $> 5 \times 10^6$ | $> 5 \times 10^6$ | $> 5 \times 10^6$ |
| BREAKOUT$_{100}$ | $3.62 \times 10^6$ | $> 5 \times 10^6$ | $> 5 \times 10^6$ | $> 5 \times 10^6$ |
| BREAKOUT$_{200}$ | $3.69 \times 10^6$ | $> 5 \times 10^6$ | $> 5 \times 10^6$ | $> 5 \times 10^6$ |
| CENTIPEDE$_{10}$ | $> 5 \times 10^6$ | $> 5 \times 10^6$ | $> 5 \times 10^6$ | $> 5 \times 10^6$ |
| CHOPPERCOMMAND$_{10}$ | $> 5 \times 10^6$ | $3.61 \times 10^6$ | $> 5 \times 10^6$ | $> 5 \times 10^6$ |
| CRAZYCLIMBER$_{20}$ | $4.00 \times 10^4$ | $3.00 \times 10^4$ | $3.00 \times 10^4$ | $> 5 \times 10^6$ |
| CRAZYCLIMBER$_{30}$ | $3.70 \times 10^5$ | $9.80 \times 10^5$ | $1.30 \times 10^5$ | $> 5 \times 10^6$ |
| DEMONATTACK$_{10}$ | $1.75 \times 10^6$ | $5.80 \times 10^5$ | $3.19 \times 10^6$ | $> 5 \times 10^6$ |
| ENDURO$_{10}$ | $4.60 \times 10^5$ | $5.00 \times 10^5$ | $> 5 \times 10^6$ | $> 5 \times 10^6$ |
| FISHINGDERBY$_{10}$ | $3.30 \times 10^5$ | $2.05 \times 10^6$ | $> 5 \times 10^6$ | $> 5 \times 10^6$ |
| FREEWAY$_{10}$ | $1.00 \times 10^4$ | $2.00 \times 10^4$ | $1.00 \times 10^4$ | $1.00 \times 10^4$ |

| | | | | |
|---|---|---|---|---|
| FREEWAY$_{20}$ | $1.00 \times 10^4$ | $1.00 \times 10^4$ | $1.00 \times 10^4$ | $5.80 \times 10^5$ |
| FREEWAY$_{30}$ | $2.10 \times 10^5$ | $4.90 \times 10^5$ | $1.08 \times 10^6$ | $> 5 \times 10^6$ |
| FREEWAY$_{40}$ | $3.20 \times 10^5$ | $6.10 \times 10^5$ | $> 5 \times 10^6$ | $> 5 \times 10^6$ |
| FREEWAY$_{50}$ | $4.80 \times 10^5$ | $7.40 \times 10^5$ | $> 5 \times 10^6$ | $> 5 \times 10^6$ |
| FREEWAY$_{70}$ | $> 5 \times 10^6$ | $3.63 \times 10^6$ | $> 5 \times 10^6$ | $> 5 \times 10^6$ |
| FREEWAY$_{100}$ | $> 5 \times 10^6$ | $> 5 \times 10^6$ | $> 5 \times 10^6$ | $> 5 \times 10^6$ |
| FREEWAY$_{200}$ | $> 5 \times 10^6$ | $> 5 \times 10^6$ | $> 5 \times 10^6$ | $> 5 \times 10^6$ |
| FROSTBITE$_{10}$ | $6.00 \times 10^4$ | $5.50 \times 10^5$ | $5.50 \times 10^5$ | $6.80 \times 10^5$ |
| GOPHER$_{30}$ | $1.10 \times 10^5$ | $1.80 \times 10^5$ | $9.00 \times 10^4$ | $1.49 \times 10^6$ |
| GOPHER$_{40}$ | $> 5 \times 10^6$ | $> 5 \times 10^6$ | $> 5 \times 10^6$ | $> 5 \times 10^6$ |
| HERO$_{10}$ | $3.00 \times 10^4$ | $3.00 \times 10^4$ | $2.00 \times 10^4$ | $1.56 \times 10^6$ |
| ICEHOCKEY$_{10}$ | $3.00 \times 10^4$ | $1.99 \times 10^6$ | $1.08 \times 10^6$ | $> 5 \times 10^6$ |
| KANGAROO$_{20}$ | $> 5 \times 10^6$ | $> 5 \times 10^6$ | $> 5 \times 10^6$ | $> 5 \times 10^6$ |
| KANGAROO$_{30}$ | $> 5 \times 10^6$ | $> 5 \times 10^6$ | $> 5 \times 10^6$ | $> 5 \times 10^6$ |
| MONTEZUMAREVENGE$_{15}$ | $> 5 \times 10^6$ | $> 5 \times 10^6$ | $> 5 \times 10^6$ | $> 5 \times 10^6$ |
| MSPACMAN$_{20}$ | $> 5 \times 10^6$ | $> 5 \times 10^6$ | $> 5 \times 10^6$ | $> 5 \times 10^6$ |
| NAMETHISGAME$_{20}$ | $2.90 \times 10^5$ | $8.30 \times 10^5$ | $1.37 \times 10^6$ | $> 5 \times 10^6$ |
| PHOENIX$_{10}$ | $8.20 \times 10^5$ | $> 5 \times 10^6$ | $> 5 \times 10^6$ | $> 5 \times 10^6$ |
| PONG$_{20}$ | $9.30 \times 10^5$ | $2.00 \times 10^5$ | $5.60 \times 10^5$ | $> 5 \times 10^6$ |
| PONG$_{30}$ | $4.70 \times 10^5$ | $6.40 \times 10^5$ | $> 5 \times 10^6$ | $> 5 \times 10^6$ |
| PONG$_{40}$ | $> 5 \times 10^6$ | $> 5 \times 10^6$ | $> 5 \times 10^6$ | $> 5 \times 10^6$ |
| PONG$_{50}$ | $> 5 \times 10^6$ | $> 5 \times 10^6$ | $> 5 \times 10^6$ | $> 5 \times 10^6$ |
| PONG$_{70}$ | $> 5 \times 10^6$ | $> 5 \times 10^6$ | $> 5 \times 10^6$ | $> 5 \times 10^6$ |
| PONG$_{100}$ | $> 5 \times 10^6$ | $> 5 \times 10^6$ | $> 5 \times 10^6$ | $> 5 \times 10^6$ |
| PRIVATEEYE$_{10}$ | $1.30 \times 10^5$ | $1.00 \times 10^5$ | $2.20 \times 10^5$ | $2.76 \times 10^6$ |
| QBERT$_{10}$ | $1.60 \times 10^5$ | $> 5 \times 10^6$ | $1.40 \times 10^5$ | $> 5 \times 10^6$ |
| QBERT$_{20}$ | $1.25 \times 10^6$ | $> 5 \times 10^6$ | $> 5 \times 10^6$ | $> 5 \times 10^6$ |
| ROADRUNNER$_{10}$ | $1.70 \times 10^5$ | $7.70 \times 10^5$ | $9.90 \times 10^5$ | $5.22 \times 10^6$ |
| SEAQUEST$_{10}$ | $1.00 \times 10^4$ | $5.00 \times 10^4$ | $1.00 \times 10^4$ | $> 5 \times 10^6$ |
| SKIING$_{10}$ | $> 5 \times 10^6$ | $> 5 \times 10^6$ | $> 5 \times 10^6$ | $> 5 \times 10^6$ |
| SPACEINVADERS$_{10}$ | $3.00 \times 10^5$ | $2.60 \times 10^5$ | $1.50 \times 10^5$ | $1.01 \times 10^6$ |
| TENNIS$_{10}$ | $> 5 \times 10^6$ | $1.48 \times 10^6$ | $> 5 \times 10^6$ | $> 5 \times 10^6$ |
| TIMEPILOT$_{10}$ | $6.00 \times 10^4$ | $5.90 \times 10^5$ | $1.50 \times 10^5$ | $4.79 \times 10^6$ |
| TUTANKHAM$_{10}$ | $2.70 \times 10^6$ | $3.81 \times 10^6$ | $2.08 \times 10^6$ | $> 5 \times 10^6$ |
| VIDEOPINBALL$_{10}$ | $> 5 \times 10^6$ | $1.70 \times 10^6$ | $1.04 \times 10^6$ | $6.90 \times 10^5$ |
| WIZARDOFWOR$_{20}$ | $> 5 \times 10^6$ | $> 5 \times 10^6$ | $> 5 \times 10^6$ | $> 5 \times 10^6$ |
| BIGFISH$_{10}^{\text{E0}}$ | $> 5 \times 10^6$ | $4.60 \times 10^5$ | $> 5 \times 10^6$ | $> 5 \times 10^6$ |
| BIGFISH$_{10}^{\text{E1}}$ | $4.00 \times 10^5$ | $2.80 \times 10^5$ | $2.70 \times 10^5$ | $> 5 \times 10^6$ |
| BIGFISH$_{10}^{\text{E2}}$ | $1.37 \times 10^6$ | $5.90 \times 10^5$ | $2.36 \times 10^6$ | $> 5 \times 10^6$ |
| BIGFISH$_{10}^{\text{H0}}$ | $> 5 \times 10^6$ | $4.30 \times 10^5$ | $1.71 \times 10^6$ | $> 5 \times 10^6$ |
| CHASER$_{20}^{\text{E0}}$ | $1.90 \times 10^5$ | $> 5 \times 10^6$ | $> 5 \times 10^6$ | $> 5 \times 10^6$ |
| CHASER$_{20}^{\text{E1}}$ | $> 5 \times 10^6$ | $> 5 \times 10^6$ | $> 5 \times 10^6$ | $> 5 \times 10^6$ |
| CHASER$_{20}^{\text{E2}}$ | $3.80 \times 10^5$ | $> 5 \times 10^6$ | $> 5 \times 10^6$ | $> 5 \times 10^6$ |
| CHASER$_{20}^{\text{H0}}$ | $1.60 \times 10^5$ | $> 5 \times 10^6$ | $> 5 \times 10^6$ | $> 5 \times 10^6$ |
| CLIMBER$_{10}^{\text{E0}}$ | $1.00 \times 10^5$ | $4.70 \times 10^5$ | $8.00 \times 10^4$ | $> 5 \times 10^6$ |
| CLIMBER$_{10}^{\text{E1}}$ | $8.00 \times 10^4$ | $> 5 \times 10^6$ | $2.00 \times 10^5$ | $1.20 \times 10^5$ |
| CLIMBER$_{10}^{\text{E2}}$ | $4.40 \times 10^5$ | $> 5 \times 10^6$ | $1.37 \times 10^6$ | $> 5 \times 10^6$ |
| CLIMBER$_{10}^{\text{H0}}$ | $> 5 \times 10^6$ | $> 5 \times 10^6$ | $> 5 \times 10^6$ | $> 5 \times 10^6$ |
| COINRUN$_{10}^{\text{E0}}$ | $> 5 \times 10^6$ | $> 5 \times 10^6$ | $> 5 \times 10^6$ | $> 5 \times 10^6$ |
| COINRUN$_{10}^{\text{E1}}$ | $3.21 \times 10^6$ | $> 5 \times 10^6$ | $1.58 \times 10^6$ | $> 5 \times 10^6$ |
| COINRUN$_{10}^{\text{E2}}$ | $> 5 \times 10^6$ | $> 5 \times 10^6$ | $> 5 \times 10^6$ | $> 5 \times 10^6$ |
| COINRUN$_{10}^{\text{H0}}$ | $1.00 \times 10^4$ | $2.00 \times 10^4$ | $2.00 \times 10^4$ | $2.80 \times 10^5$ |
| DODGEBALL$_{10}^{\text{E0}}$ | $3.70 \times 10^5$ | $2.40 \times 10^5$ | $1.57 \times 10^6$ | $> 5 \times 10^6$ |
| DODGEBALL$_{10}^{\text{E1}}$ | $6.50 \times 10^5$ | $1.42 \times 10^6$ | $2.29 \times 10^6$ | $> 5 \times 10^6$ |
| DODGEBALL$_{10}^{\text{E2}}$ | $2.70 \times 10^5$ | $2.57 \times 10^6$ | $1.60 \times 10^5$ | $6.10 \times 10^5$ |

| | | | | |
|---|---|---|---|---|
| $\text{DODGEBALL}_{10}^{\text{H0}}$ | $1.03 \times 10^6$ | $2.68 \times 10^6$ | $2.65 \times 10^6$ | $> 5 \times 10^6$ |
| $\text{FRUITBOT}_{40}^{\text{E0}}$ | $> 5 \times 10^6$ | $4.80 \times 10^5$ | $> 5 \times 10^6$ | $> 5 \times 10^6$ |
| $\text{FRUITBOT}_{40}^{\text{E1}}$ | $1.98 \times 10^6$ | $1.03 \times 10^6$ | $> 5 \times 10^6$ | $> 5 \times 10^6$ |
| $\text{FRUITBOT}_{40}^{\text{E2}}$ | $6.10 \times 10^5$ | $4.00 \times 10^4$ | $5.00 \times 10^4$ | $> 5 \times 10^6$ |
| $\text{FRUITBOT}_{40}^{\text{H0}}$ | $> 5 \times 10^6$ | $> 5 \times 10^6$ | $> 5 \times 10^6$ | $> 5 \times 10^6$ |
| $\text{HEIST}_{10}^{\text{E1}}$ | $8.10 \times 10^5$ | $1.65 \times 10^6$ | $> 5 \times 10^6$ | $> 5 \times 10^6$ |
| $\text{JUMPER}_{10}^{\text{H0}}$ | $> 5 \times 10^6$ | $> 5 \times 10^6$ | $> 5 \times 10^6$ | $> 5 \times 10^6$ |
| $\text{JUMPER}_{20}^{\text{E0}}$ | $1.00 \times 10^4$ | $> 5 \times 10^6$ | $1.00 \times 10^4$ | $1.00 \times 10^4$ |
| $\text{JUMPER}_{20}^{\text{E1}}$ | $1.00 \times 10^4$ | $> 5 \times 10^6$ | $1.00 \times 10^4$ | $1.00 \times 10^4$ |
| $\text{JUMPER}_{20}^{\text{E2}}$ | $1.90 \times 10^5$ | $> 5 \times 10^6$ | $2.60 \times 10^5$ | $> 5 \times 10^6$ |
| $\text{JUMPER}_{20}^{\text{EX}}$ | $> 5 \times 10^6$ | $> 5 \times 10^6$ | $> 5 \times 10^6$ | $> 5 \times 10^6$ |
| $\text{LEAPER}_{20}^{\text{E1}}$ | $> 5 \times 10^6$ | $> 5 \times 10^6$ | $> 5 \times 10^6$ | $> 5 \times 10^6$ |
| $\text{LEAPER}_{20}^{\text{E2}}$ | $1.07 \times 10^6$ | $> 5 \times 10^6$ | $> 5 \times 10^6$ | $> 5 \times 10^6$ |
| $\text{LEAPER}_{20}^{\text{H0}}$ | $> 5 \times 10^6$ | $> 5 \times 10^6$ | $> 5 \times 10^6$ | $> 5 \times 10^6$ |
| $\text{LEAPER}_{20}^{\text{EX}}$ | $> 5 \times 10^6$ | $> 5 \times 10^6$ | $> 5 \times 10^6$ | $> 5 \times 10^6$ |
| $\text{MAZE}_{30}^{\text{E0}}$ | $1.00 \times 10^4$ | $> 5 \times 10^6$ | $1.00 \times 10^4$ | $1.00 \times 10^4$ |
| $\text{MAZE}_{30}^{\text{E1}}$ | $> 5 \times 10^6$ | $> 5 \times 10^6$ | $> 5 \times 10^6$ | $> 5 \times 10^6$ |
| $\text{MAZE}_{30}^{\text{E2}}$ | $1.00 \times 10^4$ | $4.65 \times 10^6$ | $1.00 \times 10^4$ | $1.00 \times 10^4$ |
| $\text{MAZE}_{30}^{\text{H0}}$ | $4.28 \times 10^6$ | $> 5 \times 10^6$ | $> 5 \times 10^6$ | $> 5 \times 10^6$ |
| $\text{MAZE}_{100}^{\text{EX}}$ | $> 5 \times 10^6$ | $> 5 \times 10^6$ | $> 5 \times 10^6$ | $> 5 \times 10^6$ |
| $\text{MINER}_{10}^{\text{E0}}$ | $2.40 \times 10^5$ | $5.30 \times 10^5$ | $2.30 \times 10^5$ | $> 5 \times 10^6$ |
| $\text{MINER}_{10}^{\text{E1}}$ | $1.30 \times 10^5$ | $4.70 \times 10^5$ | $1.20 \times 10^5$ | $> 5 \times 10^6$ |
| $\text{MINER}_{10}^{\text{E2}}$ | $4.00 \times 10^4$ | $2.87 \times 10^6$ | $1.00 \times 10^4$ | $5.20 \times 10^5$ |
| $\text{MINER}_{10}^{\text{H0}}$ | $3.60 \times 10^5$ | $4.90 \times 10^5$ | $4.00 \times 10^5$ | $> 5 \times 10^6$ |
| $\text{NINJA}_{10}^{\text{E0}}$ | $9.70 \times 10^5$ | $> 5 \times 10^6$ | $> 5 \times 10^6$ | $> 5 \times 10^6$ |
| $\text{NINJA}_{10}^{\text{E1}}$ | $2.03 \times 10^6$ | $> 5 \times 10^6$ | $> 5 \times 10^6$ | $> 5 \times 10^6$ |
| $\text{NINJA}_{10}^{\text{E2}}$ | $> 5 \times 10^6$ | $> 5 \times 10^6$ | $> 5 \times 10^6$ | $> 5 \times 10^6$ |
| $\text{NINJA}_{10}^{\text{H0}}$ | $> 5 \times 10^6$ | $> 5 \times 10^6$ | $> 5 \times 10^6$ | $> 5 \times 10^6$ |
| $\text{PLUNDER}_{10}^{\text{E0}}$ | $3.00 \times 10^4$ | $2.00 \times 10^4$ | $1.00 \times 10^4$ | $1.30 \times 10^5$ |
| $\text{PLUNDER}_{10}^{\text{E1}}$ | $1.00 \times 10^4$ | $2.00 \times 10^4$ | $1.00 \times 10^4$ | $6.00 \times 10^4$ |
| $\text{PLUNDER}_{10}^{\text{E2}}$ | $1.20 \times 10^5$ | $3.80 \times 10^5$ | $4.07 \times 10^6$ | $2.25 \times 10^6$ |
| $\text{PLUNDER}_{10}^{\text{H0}}$ | $1.00 \times 10^4$ | $3.00 \times 10^4$ | $1.00 \times 10^4$ | $1.00 \times 10^5$ |
| $\text{STARPILOT}_{10}^{\text{E0}}$ | $2.65 \times 10^6$ | $6.70 \times 10^5$ | $2.22 \times 10^6$ | $> 5 \times 10^6$ |
| $\text{STARPILOT}_{10}^{\text{E1}}$ | $6.00 \times 10^5$ | $8.30 \times 10^5$ | $1.14 \times 10^6$ | $> 5 \times 10^6$ |
| $\text{STARPILOT}_{10}^{\text{E2}}$ | $8.40 \times 10^5$ | $1.43 \times 10^6$ | $3.97 \times 10^6$ | $> 5 \times 10^6$ |
| $\text{STARPILOT}_{10}^{\text{H0}}$ | $> 5 \times 10^6$ | $2.52 \times 10^6$ | $> 5 \times 10^6$ | $> 5 \times 10^6$ |
| $\text{EMPTY-5X5}$ | $1.40 \times 10^5$ | $3.90 \times 10^5$ | $3.90 \times 10^5$ | $1.99 \times 10^6$ |
| $\text{EMPTY-6X6}$ | $4.00 \times 10^5$ | $3.10 \times 10^5$ | $5.50 \times 10^5$ | $> 5 \times 10^6$ |
| $\text{EMPTY-8X8}$ | $3.40 \times 10^5$ | $3.60 \times 10^5$ | $9.10 \times 10^5$ | $> 5 \times 10^6$ |
| $\text{EMPTY-16X16}$ | $1.01 \times 10^6$ | $> 5 \times 10^6$ | $> 5 \times 10^6$ | $> 5 \times 10^6$ |
| $\text{DOORKEY-5X5}$ | $5.90 \times 10^5$ | $7.90 \times 10^5$ | $3.22 \times 10^6$ | $> 5 \times 10^6$ |
| $\text{DOORKEY-6X6}$ | $1.67 \times 10^6$ | $> 5 \times 10^6$ | $> 5 \times 10^6$ | $> 5 \times 10^6$ |
| $\text{DOORKEY-8X8}$ | $> 5 \times 10^6$ | $> 5 \times 10^6$ | $> 5 \times 10^6$ | $> 5 \times 10^6$ |
| $\text{DOORKEY-16X16}$ | $> 5 \times 10^6$ | $> 5 \times 10^6$ | $> 5 \times 10^6$ | $> 5 \times 10^6$ |
| $\text{MULTIROOM-N2-S4}$ | $6.10 \times 10^5$ | $9.80 \times 10^5$ | $3.81 \times 10^6$ | $> 5 \times 10^6$ |
| $\text{MULTIROOM-N4-S5}$ | $> 5 \times 10^6$ | $> 5 \times 10^6$ | $> 5 \times 10^6$ | $> 5 \times 10^6$ |
| $\text{MULTIROOM-N6}$ | $> 5 \times 10^6$ | $> 5 \times 10^6$ | $> 5 \times 10^6$ | $> 5 \times 10^6$ |
| $\text{KEYCORRIDORS3R1}$ | $1.40 \times 10^6$ | $> 5 \times 10^6$ | $> 5 \times 10^6$ | $> 5 \times 10^6$ |
| $\text{KEYCORRIDORS3R2}$ | $> 5 \times 10^6$ | $> 5 \times 10^6$ | $> 5 \times 10^6$ | $> 5 \times 10^6$ |
| $\text{KEYCORRIDORS3R3}$ | $> 5 \times 10^6$ | $> 5 \times 10^6$ | $> 5 \times 10^6$ | $> 5 \times 10^6$ |
| $\text{KEYCORRIDORS4R3}$ | $> 5 \times 10^6$ | $> 5 \times 10^6$ | $> 5 \times 10^6$ | $> 5 \times 10^6$ |
| $\text{UNLOCK}$ | $1.23 \times 10^6$ | $> 5 \times 10^6$ | $> 5 \times 10^6$ | $> 5 \times 10^6$ |
| $\text{UNLOCKPICKUP}$ | $> 5 \times 10^6$ | $> 5 \times 10^6$ | $> 5 \times 10^6$ | $> 5 \times 10^6$ |
| $\text{BLOCKEDUNLOCKPICKUP}$ | $> 5 \times 10^6$ | $> 5 \times 10^6$ | $> 5 \times 10^6$ | $> 5 \times 10^6$ |

| | | | | |
|---|---|---|---|---|
| OBSTRUCTEDMAZE-1DL | $> 5 \times 10^6$ | $> 5 \times 10^6$ | $> 5 \times 10^6$ | $> 5 \times 10^6$ |
| OBSTRUCTEDMAZE-1DLH | $> 5 \times 10^6$ | $> 5 \times 10^6$ | $> 5 \times 10^6$ | $> 5 \times 10^6$ |
| OBSTRUCTEDMAZE-1DLHB | $> 5 \times 10^6$ | $> 5 \times 10^6$ | $> 5 \times 10^6$ | $> 5 \times 10^6$ |
| FOURROOMS | $> 5 \times 10^6$ | $> 5 \times 10^6$ | $> 5 \times 10^6$ | $> 5 \times 10^6$ |
| LAVACROSSINGS9N1 | $7.10 \times 10^5$ | $4.60 \times 10^5$ | $2.78 \times 10^6$ | $> 5 \times 10^6$ |
| LAVACROSSINGS9N2 | $2.38 \times 10^6$ | $4.60 \times 10^5$ | $3.17 \times 10^6$ | $> 5 \times 10^6$ |
| LAVACROSSINGS9N3 | $6.20 \times 10^5$ | $6.40 \times 10^5$ | $2.02 \times 10^6$ | $> 5 \times 10^6$ |
| LAVACROSSINGS11N5 | $> 5 \times 10^6$ | $> 5 \times 10^6$ | $> 5 \times 10^6$ | $> 5 \times 10^6$ |
| SIMPLECROSSINGS9N1 | $4.50 \times 10^5$ | $8.70 \times 10^5$ | $> 5 \times 10^6$ | $> 5 \times 10^6$ |
| SIMPLECROSSINGS9N2 | $5.10 \times 10^5$ | $1.94 \times 10^6$ | $> 5 \times 10^6$ | $> 5 \times 10^6$ |
| SIMPLECROSSINGS9N3 | $3.60 \times 10^5$ | $5.00 \times 10^5$ | $1.18 \times 10^6$ | $> 5 \times 10^6$ |
| SIMPLECROSSINGS11N5 | $> 5 \times 10^6$ | $> 5 \times 10^6$ | $> 5 \times 10^6$ | $> 5 \times 10^6$ |
| LAVAGAPS5 | $2.70 \times 10^5$ | $2.80 \times 10^5$ | $7.10 \times 10^5$ | $> 5 \times 10^6$ |
| LAVAGAPS6 | $9.70 \times 10^5$ | $6.30 \times 10^5$ | $2.44 \times 10^6$ | $> 5 \times 10^6$ |
| LAVAGAPS7 | $1.26 \times 10^6$ | $1.25 \times 10^6$ | $> 5 \times 10^6$ | $> 5 \times 10^6$ |

### D.2 TABLE OF RETURNS

This table lists the optimal returns in each sticky-action MDP as well as the highest returns achieved by PPO, DQN, SQIRL, and GORP. The achieved returns may be higher than the optimal return because they are measured by a Monte Carlo average over 100 episodes during evaluation.

| | | Returns | | |
|---|---|---|---|---|
| MDP | Optimal policy | PPO | DQN | SQIRL |
| ALIEN$_{10}$ | 158.13 | 158.1 | 157.2 | 155.7 |
| AMIDAR$_{20}$ | 76.49 | 63.54 | 71.44 | 60.07 |
| ASSAULT$_{10}$ | 105.0 | 105.0 | 105.0 | 105.0 |
| ASTERIX$_{10}$ | 327.53 | 330.0 | 334.5 | 332.5 |
| ASTEROIDS$_{10}$ | 170.81 | 138.1 | 115.3 | 130.1 |
| ATLANTIS$_{10}$ | 187.5 | 190.0 | 190.0 | 192.0 |
| ATLANTIS$_{20}$ | 740.91 | 752.0 | 726.0 | 754.0 |
| ATLANTIS$_{30}$ | 1,829.52 | 1,238.0 | 995.0 | 1,117.0 |
| ATLANTIS$_{40}$ | 2,620.35 | 1,849.0 | 1,218.0 | 1,677.0 |
| ATLANTIS$_{50}$ | 4,856.81 | 3,683.0 | 2,059.0 | 3,140.0 |
| ATLANTIS$_{70}$ | 7,932.88 | 6,051.0 | 3,222.0 | 5,456.0 |
| BANKHEIST$_{10}$ | 26.15 | 26.6 | 26.4 | 26.2 |
| BATTLEZONE$_{10}$ | 1,497.07 | 1,550.0 | 1,520.0 | 1,560.0 |
| BEAMRIDER$_{20}$ | 129.23 | 125.4 | 129.36 | 124.08 |
| BOWLING$_{30}$ | 8.8 | 8.81 | 5.75 | 7.69 |
| BREAKOUT$_{10}$ | 1.17 | 1.26 | 1.21 | 1.25 |
| BREAKOUT$_{20}$ | 1.93 | 1.97 | 1.99 | 2.02 |
| BREAKOUT$_{30}$ | 2.5 | 2.55 | 2.25 | 2.49 |
| BREAKOUT$_{40}$ | 2.61 | 2.7 | 2.34 | 2.49 |
| BREAKOUT$_{50}$ | 2.69 | 2.84 | 2.42 | 2.47 |
| BREAKOUT$_{70}$ | 2.9 | 3.01 | 2.38 | 2.53 |
| BREAKOUT$_{100}$ | 3.08 | 3.12 | 0.81 | 2.44 |
| BREAKOUT$_{200}$ | 3.08 | 3.08 | 0.78 | 2.5 |
| CENTIPEDE$_{10}$ | 1,321.17 | 900.0 | 1,150.5 | 1,186.61 |
| CHOPPERCOMMAND$_{10}$ | 553.42 | 469.0 | 560.0 | 495.0 |
| CRAZYCLIMBER$_{20}$ | 324.9 | 328.0 | 327.0 | 333.0 |
| CRAZYCLIMBER$_{30}$ | 698.07 | 701.0 | 704.0 | 707.0 |
| DEMONATTACK$_{10}$ | 37.07 | 37.1 | 38.3 | 37.3 |
| ENDURO$_{10}$ | 4.27 | 4.33 | 4.34 | 3.95 |
| FISHINGDERBY$_{10}$ | 7.5 | 7.56 | 7.5 | 6.79 |
| FREEWAY$_{10}$ | 1.0 | 1.0 | 1.0 | 1.0 |
| FREEWAY$_{20}$ | 2.0 | 2.0 | 2.0 | 2.0 |
| FREEWAY$_{30}$ | 3.75 | 3.77 | 3.77 | 3.79 |

| | | | | |
|---|---:|---:|---:|---:|
| $\text{FREEWAY}_{40}$ | 4.75 | 4.78 | 4.76 | 4.69 |
| $\text{FREEWAY}_{50}$ | 5.75 | 5.78 | 5.76 | 5.67 |
| $\text{FREEWAY}_{70}$ | 8.49 | 7.84 | 8.5 | 8.03 |
| $\text{FREEWAY}_{100}$ | 11.83 | 10.85 | 11.47 | 10.86 |
| $\text{FREEWAY}_{200}$ | 23.84 | 21.53 | 22.05 | 20.88 |
| $\text{FROSTBITE}_{10}$ | 66.72 | 67.5 | 67.2 | 67.5 |
| $\text{GOPHER}_{30}$ | 18.75 | 19.2 | 19.0 | 19.4 |
| $\text{GOPHER}_{40}$ | 112.93 | 83.4 | 72.6 | 72.8 |
| $\text{HERO}_{10}$ | 74.71 | 75.0 | 75.0 | 75.0 |
| $\text{ICEHOCKEY}_{10}$ | 1.0 | 1.0 | 1.0 | 1.0 |
| $\text{KANGAROO}_{20}$ | 186.32 | 174.0 | 180.0 | 168.0 |
| $\text{KANGAROO}_{30}$ | 444.7 | 200.0 | 298.0 | 207.0 |
| $\text{MONTEZUMAREVENGE}_{15}$ | 22.53 | 0.0 | 0.0 | 0.0 |
| $\text{MSPACMAN}_{20}$ | 460.6 | 282.9 | 435.5 | 420.0 |
| $\text{NAMETHISGAME}_{20}$ | 94.02 | 95.7 | 96.3 | 94.8 |
| $\text{PHOENIX}_{10}$ | 179.89 | 180.8 | 173.8 | 169.2 |
| $\text{PONG}_{20}$ | $-1.01$ | $-1.0$ | $-1.0$ | $-1.0$ |
| $\text{PONG}_{30}$ | $-1.61$ | $-1.59$ | $-1.59$ | $-1.85$ |
| $\text{PONG}_{40}$ | $-1.11$ | $-1.26$ | $-1.26$ | $-2.0$ |
| $\text{PONG}_{50}$ | $-1.36$ | $-1.96$ | $-1.52$ | $-3.35$ |
| $\text{PONG}_{70}$ | $-1.48$ | $-3.43$ | $-2.32$ | $-5.23$ |
| $\text{PONG}_{100}$ | $-1.41$ | $-5.11$ | $-4.18$ | $-9.05$ |
| $\text{PRIVATEEYE}_{10}$ | 98.44 | 99.0 | 99.0 | 99.0 |
| $\text{QBERT}_{10}$ | 350.0 | 351.75 | 339.75 | 361.75 |
| $\text{QBERT}_{20}$ | 579.71 | 582.75 | 565.0 | 547.0 |
| $\text{ROADRUNNER}_{10}$ | 474.71 | 489.0 | 486.0 | 486.0 |
| $\text{SEAQUEST}_{10}$ | 20.0 | 20.0 | 20.0 | 20.0 |
| $\text{SKIING}_{10}$ | $-8,011.57$ | $-9,013.0$ | $-8,353.5$ | $-8,341.18$ |
| $\text{SPACEINVADERS}_{10}$ | 33.91 | 34.3 | 34.8 | 34.8 |
| $\text{TENNIS}_{10}$ | 0.8 | 0.0 | 0.81 | 0.59 |
| $\text{TIMEPILOT}_{10}$ | 131.25 | 136.0 | 135.0 | 138.0 |
| $\text{TUTANKHAM}_{10}$ | 16.51 | 16.79 | 16.54 | 16.83 |
| $\text{VIDEOPINBALL}_{10}$ | 1,744.95 | 1,388.09 | 1,816.01 | 1,825.03 |
| $\text{WIZARDOFWOR}_{20}$ | 260.35 | 100.0 | 100.0 | 100.0 |
| $\text{BIGFISH}_{10}^{\text{E0}}$ | 3.53 | 2.26 | 3.56 | 2.78 |
| $\text{BIGFISH}_{10}^{\text{E1}}$ | 2.97 | 2.98 | 2.98 | 2.99 |
| $\text{BIGFISH}_{10}^{\text{E2}}$ | 6.86 | 6.87 | 6.89 | 6.87 |
| $\text{BIGFISH}_{10}^{\text{H0}}$ | 2.59 | 1.84 | 2.62 | 2.64 |
| $\text{CHASER}_{20}^{\text{E0}}$ | 0.88 | 0.88 | 0.4168 | 0.8052 |
| $\text{CHASER}_{20}^{\text{E1}}$ | 0.88 | 0.84 | 0.8 | 0.8628 |
| $\text{CHASER}_{20}^{\text{E2}}$ | 0.88 | 0.8748 | 0.5304 | 0.8424 |
| $\text{CHASER}_{20}^{\text{H0}}$ | 0.88 | 0.8792 | 0.8572 | 0.84 |
| $\text{CLIMBER}_{10}^{\text{E0}}$ | 1.93 | 1.94 | 1.94 | 1.95 |
| $\text{CLIMBER}_{10}^{\text{E1}}$ | 1.75 | 1.78 | 1.62 | 1.78 |
| $\text{CLIMBER}_{10}^{\text{E2}}$ | 10.92 | 11.0 | 10.89 | 11.0 |
| $\text{CLIMBER}_{10}^{\text{H0}}$ | 1.22 | 1.0 | 1.0 | 1.0 |
| $\text{COINRUN}_{10}^{\text{E0}}$ | 8.82 | 8.8 | 7.0 | 8.8 |
| $\text{COINRUN}_{10}^{\text{E1}}$ | 8.36 | 8.4 | 7.3 | 8.6 |
| $\text{COINRUN}_{10}^{\text{E2}}$ | 6.94 | 6.9 | 6.0 | 0.0 |
| $\text{COINRUN}_{10}^{\text{H0}}$ | 10.0 | 10.0 | 10.0 | 10.0 |
| $\text{DODGEBALL}_{10}^{\text{E0}}$ | 7.81 | 8.08 | 8.12 | 8.16 |
| $\text{DODGEBALL}_{10}^{\text{E1}}$ | 5.3 | 5.32 | 5.36 | 5.3 |
| $\text{DODGEBALL}_{10}^{\text{E2}}$ | 5.71 | 5.82 | 5.78 | 5.88 |
| $\text{DODGEBALL}_{10}^{\text{H0}}$ | 4.13 | 4.28 | 4.16 | 4.2 |
| $\text{FRUITBOT}_{40}^{\text{E0}}$ | 1.99 | 1.33 | 2.24 | 1.35 |
| $\text{FRUITBOT}_{40}^{\text{E1}}$ | 3.6 | 3.64 | 3.66 | 2.47 |

| | | | | |
|---|---|---|---|---|
| $\text{FRUITBOT}_{40}^{E2}$ | 0.89 | 0.91 | 0.89 | 0.91 |
| $\text{FRUITBOT}_{40}^{H0}$ | 1.85 | 0.0 | 0.11 | 0.04 |
| $\text{HEIST}_{10}^{E1}$ | 9.38 | 9.5 | 9.4 | 0.0 |
| $\text{JUMPER}_{10}^{H0}$ | 1.33 | 0.0 | 0.0 | 0.0 |
| $\text{JUMPER}_{20}^{E0}$ | 10.0 | 10.0 | 7.5 | 10.0 |
| $\text{JUMPER}_{20}^{E1}$ | 10.0 | 10.0 | 6.0 | 10.0 |
| $\text{JUMPER}_{20}^{E2}$ | 10.0 | 10.0 | 6.8 | 10.0 |
| $\text{JUMPER}_{20}^{EX}$ | 2.77 | 0.0 | 0.0 | 0.0 |
| $\text{LEAPER}_{20}^{E1}$ | 4.92 | 0.0 | 0.0 | 0.0 |
| $\text{LEAPER}_{20}^{E2}$ | 9.92 | 10.0 | 9.9 | 9.5 |
| $\text{LEAPER}_{20}^{H0}$ | 6.42 | 0.0 | 0.0 | 0.0 |
| $\text{LEAPER}_{20}^{EX}$ | 5.7 | 0.0 | 0.0 | 0.0 |
| $\text{MAZE}_{30}^{E0}$ | 10.0 | 10.0 | 0.0 | 10.0 |
| $\text{MAZE}_{30}^{E1}$ | 7.42 | 0.0 | 0.0 | 0.0 |
| $\text{MAZE}_{30}^{E2}$ | 10.0 | 10.0 | 10.0 | 10.0 |
| $\text{MAZE}_{30}^{H0}$ | 9.99 | 10.0 | 0.0 | 0.1 |
| $\text{MAZE}_{100}^{EX}$ | 9.76 | 0.0 | 0.0 | 0.0 |
| $\text{MINER}_{10}^{E0}$ | 0.91 | 0.92 | 0.91 | 0.92 |
| $\text{MINER}_{10}^{E1}$ | 1.63 | 1.69 | 1.66 | 1.68 |
| $\text{MINER}_{10}^{E2}$ | 1.0 | 1.0 | 1.0 | 1.0 |
| $\text{MINER}_{10}^{H0}$ | 2.97 | 2.97 | 2.98 | 2.99 |
| $\text{NINJA}_{10}^{E0}$ | 6.53 | 6.6 | 0.0 | 3.3 |
| $\text{NINJA}_{10}^{E1}$ | 6.08 | 6.3 | 0.0 | 0.0 |
| $\text{NINJA}_{10}^{E2}$ | 2.0 | 0.0 | 0.0 | 0.0 |
| $\text{NINJA}_{10}^{H0}$ | 2.22 | 0.0 | 0.0 | 0.0 |
| $\text{PLUNDER}_{10}^{E0}$ | 1.0 | 1.0 | 1.0 | 1.0 |
| $\text{PLUNDER}_{10}^{E1}$ | 1.0 | 1.0 | 1.0 | 1.0 |
| $\text{PLUNDER}_{10}^{E2}$ | 0.56 | 0.62 | 0.57 | 0.56 |
| $\text{PLUNDER}_{10}^{H0}$ | 1.0 | 1.0 | 1.0 | 1.0 |
| $\text{STARPILOT}_{10}^{E0}$ | 7.02 | 7.09 | 7.11 | 7.04 |
| $\text{STARPILOT}_{10}^{E1}$ | 4.33 | 4.37 | 4.37 | 4.37 |
| $\text{STARPILOT}_{10}^{E2}$ | 3.58 | 3.62 | 3.62 | 3.63 |
| $\text{STARPILOT}_{10}^{H0}$ | 3.38 | 3.33 | 3.42 | 3.27 |
| EMPTY-5X5 | 1.0 | 1.0 | 1.0 | 1.0 |
| EMPTY-6X6 | 1.0 | 1.0 | 1.0 | 1.0 |
| EMPTY-8X8 | 1.0 | 1.0 | 1.0 | 1.0 |
| EMPTY-16X16 | 1.0 | 1.0 | 0.0 | 0.1 |
| DOORKEY-5X5 | 1.0 | 1.0 | 1.0 | 1.0 |
| DOORKEY-6X6 | 1.0 | 1.0 | 0.0 | 0.03 |
| DOORKEY-8X8 | 1.0 | 0.0 | 0.0 | 0.0 |
| DOORKEY-16X16 | 1.0 | 0.0 | 0.0 | 0.0 |
| MULTIROOM-N2-S4 | 1.0 | 1.0 | 1.0 | 1.0 |
| MULTIROOM-N4-S5 | 1.0 | 0.0 | 0.0 | 0.0 |
| MULTIROOM-N6 | 1.0 | 0.0 | 0.0 | 0.0 |
| KEYCORRIDORS3R1 | 1.0 | 1.0 | 0.0 | 0.03 |
| KEYCORRIDORS3R2 | 1.0 | 0.0 | 0.0 | 0.01 |
| KEYCORRIDORS3R3 | 1.0 | 0.0 | 0.0 | 0.0 |
| KEYCORRIDORS4R3 | 1.0 | 0.0 | 0.0 | 0.0 |
| UNLOCK | 1.0 | 1.0 | 0.0 | 0.06 |
| UNLOCKPICKUP | 1.0 | 0.0 | 0.0 | 0.01 |
| BLOCKEDUNLOCKPICKUP | 1.0 | 0.0 | 0.0 | 0.0 |
| OBSTRUCTEDMAZE-1DL | 1.0 | 0.0 | 0.0 | 0.01 |
| OBSTRUCTEDMAZE-1DLH | 1.0 | 0.0 | 0.0 | 0.01 |
| OBSTRUCTEDMAZE-1DLHB | 1.0 | 0.0 | 0.0 | 0.0 |
| FOURROOMS | 1.0 | 0.0 | 0.0 | 0.08 |

| | | | | |
|---|---|---|---|---|
| LAVACROSSINGS9N1 | 1.0 | 1.0 | 1.0 | 1.0 |
| LAVACROSSINGS9N2 | 1.0 | 1.0 | 1.0 | 1.0 |
| LAVACROSSINGS9N3 | 1.0 | 1.0 | 1.0 | 1.0 |
| LAVACROSSINGS11N5 | 0.41 | 0.0 | 0.0 | 0.0 |
| SIMPLECROSSINGS9N1 | 1.0 | 1.0 | 1.0 | 0.86 |
| SIMPLECROSSINGS9N2 | 1.0 | 1.0 | 1.0 | 0.52 |
| SIMPLECROSSINGS9N3 | 1.0 | 1.0 | 1.0 | 1.0 |
| SIMPLECROSSINGS11N5 | 1.0 | 0.0 | 0.0 | 0.01 |
| LAVAGAPS5 | 1.0 | 1.0 | 1.0 | 1.0 |
| LAVAGAPS6 | 1.0 | 1.0 | 1.0 | 1.0 |
| LAVAGAPS7 | 1.0 | 1.0 | 1.0 | 0.92 |

## D.3 LEARNING CURVES FOR STICKY-ACTION BRIDGE MDPs

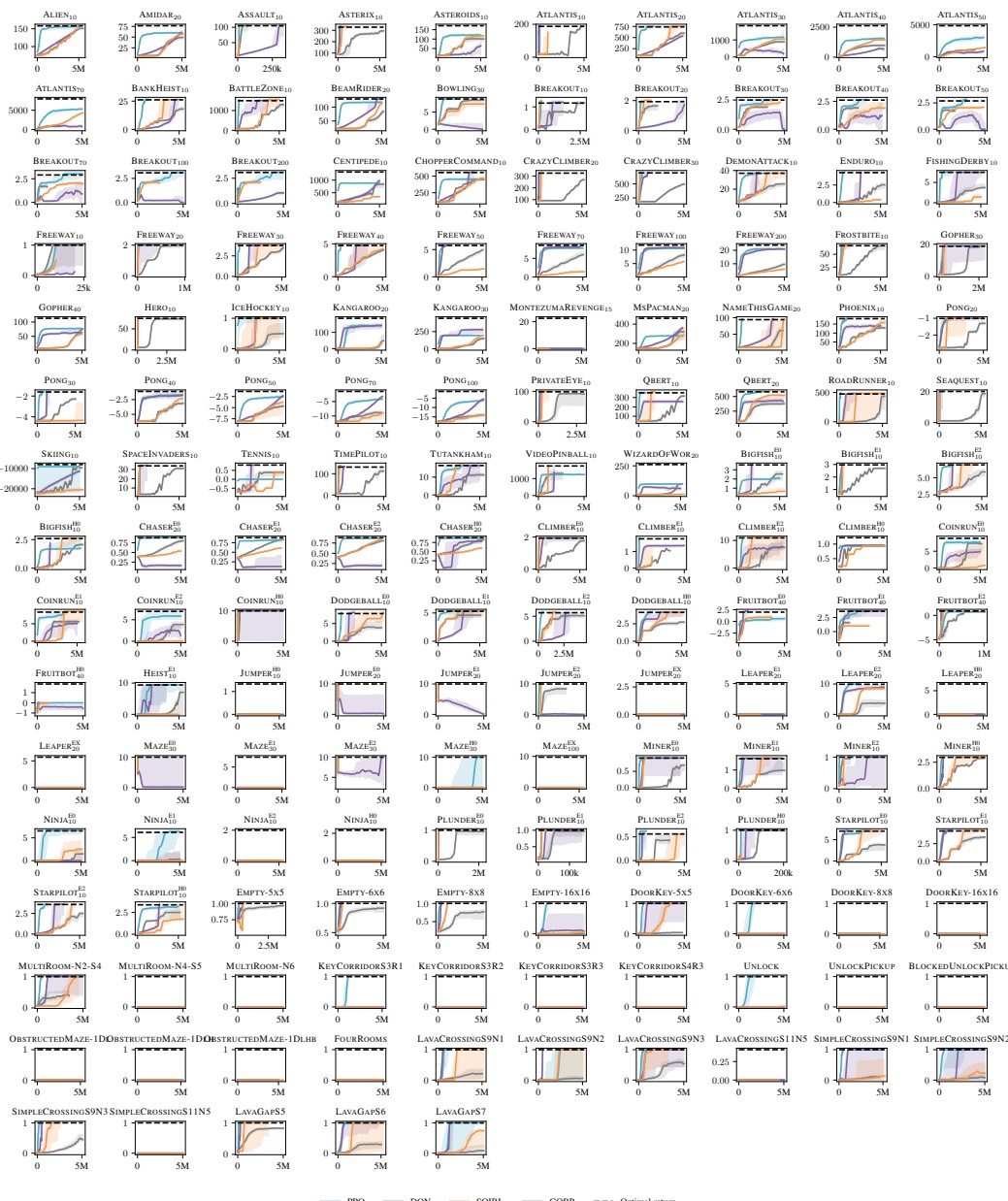

Figure 5: Learning curves for PPO, DQN, SQIRL, and SQIRL on the sticky-action BRIDGE MDPs. Solid lines show the median return (over 5 random seeds) of the policies learned by each algorithm throughout training. The shaded region shows the range of returns over random seeds. The optimal return in each environment is shown as the dashed black line.

