# OpenReview forum: "The Effective Horizon Explains Deep RL Performance in Stochastic Environments"
_ICLR.cc/2024/Conference — ICLR 2024 spotlight_

### Official Review · Reviewer_UBNV · 2023-10-25

**Soundness:** 3 good
**Presentation:** 4 excellent
**Contribution:** 3 good
**Rating:** 6
**Confidence:** 4

**Summary:**

This paper proposed a new explanation of the success of deep RL algorithms that use random exploration strategies for stochastic environments. The key observation is that many environments can be solved by a few steps of value iteration starting with the uniformly random policy, meaning that sophisticated exploration is unnecessary. Based on the observation, this paper designs a novel provable RL algorithm called SQIRL that runs a few steps of fitted-Q iteration, and the sample complexity of SQIRL is only exponential in (roughly) the number of value iteration steps needed to solve the environment. This paper also shows empirically that the sample complexity of SQIRL correlates with the sample complexity of standard deep RL algorithms such as DQN and PPO.

**Strengths:**

-	This paper extends the effective horizon definition in Laidlaw et al. (2023) to stochastic environments, and consequently designs a novel provable algorithm SQIRL that depends exponentially only on the effective horizon. This paper also empirically verifies that the effective horizon is small for many realistic environments including Atari games. Hence, the SQIRL algorithm is the first algorithm that (a) has provable sample complexity upper bound on realistic environments, and (b) achieves non-trivial performance empirically.
-	Although the algorithm is very similar to the classic FQI algorithm, the sample complexity upper bound in Theorem 3.6 is novel. In addition, the assumptions on the oracle are mild and potentially can be satisfied by realistic neural networks.
-	This paper is well-written and easy-to-follow.

**Weaknesses:**

-	Given that SQIRL has a provable and also computable sample complexity bound, this paper could benefit from more analysis on the comparison between the theoretical and empirical performance of SQIRL. For example, how does the percentage of environments that SQIRL can solve change with k? Is there a strong correlation between the actual sample complexity SQIRL and the sample complexity upper bound in Theorem 3.6?
-	For an unknown environment, computing the effective horizon or the sample complexity bound requires running a few steps of value iteration, which is not much easier than running the algorithm since we must iterate through the entire state space. This makes the instance-dependent complexity upper bound less helpful in determining the hardness of an environment / predicting the performance of deep RL algorithms on new environments. The bound could be more impactful if there is an efficient algorithm to estimate the effective horizon.
-	It is unclear whether the small effective horizon is an artifact of the sticky actions modification to the environment. A 25% chance of sticky action could potentially make long-term planning impossible, hence decreasing the effective horizon. Hence the conclusion of Figure 2 needs further justification.

**Questions:**

-	How does the performance of SQIRL change with the hyperparameter k? Is the best performance always achieved by choosing the smallest k such that the environment is k-QVI-solvable?
-	Is there a typo in the legend of Figure 2? Currently the figure shows that PPO succeeds less often when k=1.
-	It would be intriguing to see whether the sample complexity of the empirical version of SQIRL is still upper bounded by Theorem 3.6. For example, for all the environments with k=1, does SQIRL always solve the environment after $N^{SQIRL}$ steps?

---

> ### Author Response · Authors · 2023-11-16
> **Author response**
>
> We thank the reviewer for their insightful comments. We appreciate they found our results to be a significant contribution and our paper to be "well-written and easy-to-follow." The weaknesses and questions in the review are addressed below:
>  * **Comparing the theoretical and empirical performance of SQIRL:** we agree that it would be ideal to compare our theoretical bounds on SQIRL's sample complexity to its empirical performance. However, this is actually quite difficult because our practical implementation of SQIRL uses neural networks and neural network generalization is not well understood. In the bound on SQIRL's sample complexity in equation (1), we can easily calculate the effective horizon $\bar{H}_k$, but the $D$ term depends on how well the regression oracle—neural networks in our practical implementation—can generalize given a certain number of training samples. Since there are few or no bounds on neural network generalization error that reflect their empirical performance, we cannot provide a tight bound on SQIRL's sample complexity using neural networks. However, an advantage of our work is that any advances in understanding neural network generalization can immediately result in better bounds on SQIRL's sample complexity. Furthermore, we believe that the fact that SQIRL performs well empirically suggests our assumptions are met in practice and that our theory is a valuable tool for analyzing model-free RL with random exploration.
>  * **Computing the effective horizon is not much easier than solving the environment:** this is a valid point, and we agree that it is intractable to compute the effective horizon for many problems. However, we still think that it can provide useful insights to theorists and practitioners. There may be ways of bounding the effective horizon using simpler quantities. For instance, Theorem B.4 of Laidlaw et al. (2023) shows that for any deterministic goal-based environment, the effective horizon can be bounded based on the probability of reaching the goal under the random policy from various states; their analysis translates easily to the stochastic setting. It is often easy to reason about the probability of reaching a goal, making the effective horizon a useful tool for analysis in goal-based environments. Besides precise bounds like this theorem, the effective horizon also has an intuitive interpretation—when the Q-values of optimal actions under the random policy are easier to distinguish from those of suboptimal actions (i.e., the $k$-gap is larger), then it will be easier to learn an optimal policy. This type of intuitive insight can help practitioners design better environments and reward functions such that RL will easily succeed.
>  * **How does the performance of SQIRL change based on $k$?** In the table below, we show how many environments can be solved by SQIRL as we allow just $k = 1$, $k \leq 2$, or $k \leq 3$. We find that even with $k = 1$, SQIRL can solve many of the sticky-action BRIDGE environments.
>
>    |            | Number of environments solved |
>    |-|-|
>    | $k = 1$    | 48 |
>    | $k \leq 2$ | 64 |
> 	 | $k \leq 3$ | 68 |
>
>    We also found that the ideal value of $k$ for SQIRL varied depending on the environment (see the table below), and was not always the same as the minimum value of $k$ for which the MDP is approximately $k$-QVI-solvable (which for most environments was $k = 1$—see Figure 2). The reason for this is that while increasing $k$ increases the first term in the definition of the stochastic effective horizon (Definition 3.3), it might decrease the second term by making the $k$-gap larger, thus making the stochastic effective horizon $\bar{H}_k$ smaller as $k$ increases.
>
>    | Optimal value of $k$ for SQIRL | Number of environments |
>    |-|-|
>    | 1 | 19 |
>    | 2 | 22 |
>    | 3 | 27 |
>
>    We will include these tables and analysis in the final paper if accepted.
>
>  * **Is the small effective horizon just a product of sticky actions?** We believe that the low effective horizon in our environments is more of an intrinsic property that does not just result from using sticky actions, since Laidlaw et al. (2023) also found that the (deterministic) effective horizon was low in the BRIDGE environments even without sticky actions. Furthermore, sticky actions are a standard method for making environments stochastic and widely used as a way of evaluating RL algorithms [1]. Thus, the fact that the stochastic effective horizon is low in these environments suggests that it underlies the success of many deep RL algorithms in this common evaluation paradigm of sticky actions.
>  * **Is there a typo in the legend of Figure 2?** Yes, this is a typo, and we have fixed it in an updated revision of the paper.
>
> [1] Machado et al. Revisiting the arcade learning environment: Evaluation protocols and open problems for general agents. Journal of Artificial Intelligence Research, 2018.

---

> > ### Comment · Reviewer_UBNV · 2023-11-21
> >
> > Thank you for the response. I will keep my score as it is.

---

### Official Review · Reviewer_1dWJ · 2023-10-27

**Soundness:** 3 good
**Presentation:** 3 good
**Contribution:** 2 fair
**Rating:** 6
**Confidence:** 3

**Summary:**

In this work, the authors aim to provide theoretical explanations, supported by an empirical analysis, of the practical success of Deep-RL algorithms. In recent work, Laidlaw et al., 2023, "Bridging RL Theory and Practice with the Effective Horizon", the authors have shown that deep-RL methods succeed in (deterministic) MDPs, especially in cases in which it is sufficient to take few "greedyfication" steps from the Q-function of the random policy (i.e., the effective horizon). In this work, the authors show that these claims can be extended to stochastic MDPs by introducing the "stochastic effective horizon". More specifically, the paper is articulated in the following points:
1. The authors provide a formal definition of the "stochastic effective horizon" notion that presents itself as the natural extension of the effective horizon introduced by Laidlaw et al., 2023.
2. Secondly, the authors propose a simple algorithm (SQIRL) whose sample complexity scales exponentially with the stochastic effective horizon (which is typically much smaller than the optimization horizon of the underlying problem).
3. On a stochastic extension of the benchmark Bridge (Laidlaw et al., 2023), the authors show that modern deep-RL algorithms performance significantly correlates with the empirical success of SQIRL. This empirical phenomenon suggests that the concept of stochastic effective horizon can explain some of the reasons behind the success/failure of modern deep-RL methods.

**Strengths:**

1. Comprehending the factors contributing to the successes and shortcomings of deep reinforcement learning (Deep-RL) algorithms is of utmost importance. Analyzing the environments in which current methods excel provides insight into both their capabilities and constraints. This understanding serves as a foundation for inspiring and creating innovative approaches that address the limitations of current technologies. In this sense, the work done by the authors goes in this direction, as the problem deserves attention from the community (both practitioners and theoreticians).
2. The paper takes large inspiration from the recent work of Laidlaw et al., 2023, "Bridging RL Theory and Practice with the Effective Horizon". Nevertheless, in Laidlaw et al., 2023, the authors consider a deterministic setting. In this work, instead, the authors propose an extension to the more general (and challenging) stochastic environments. Although, in this sense, the novelty of the underlying idea is incremental, the challenges that are introduced from stochastic environments do not allow for a direct extension.
3. The main text is overall well-written and easy to understand.

**Weaknesses:**

**1. Novelty**

First, I remark that the main contribution done by the authors goes into the direction of providing explanations behind the success of deep-RL algorithms. Although these explanations are highly appreciated, it has to be remarked that the main idea on which the work is built has been already proposed in Laidlaw et al., 2023. Indeed, in Laidlaw et al., 2023, the authors have shown that deep-RL succeed in (deterministic) MDPs especially in cases in which it is sufficient to take few "greedyfication" steps from the Q-function of the random policy. In this sense, the novelty, in terms of new explanations that are given to the success of deep-RL algorithms is somehow limited. The contributions of the authors, in this sense, is limited to to the extension to stochastic environments.

**2. Theoretical claims and analysis**

I have concerns regarding the theoretical claims done by the authors, especially regarding the main Theorem (i.e., Theorem 3.6). Specifically, I checked the proofs behind Theorem 3.6, and there are some steps that are unclear/unprecise/uncorrect (p.s., I haven't had a look in details to the other sections of the appendix, so I am unaware if there mistakes in those parts).

First, I begin with the result on the sample complexity (Eq. 1). Using standard tools from the bandit community, it is possible to show that, for $\epsilon$-best arm identification with 2 arms, the Lower bound is given by $\widetilde{\Omega}[ \max \left( \Delta^{-2}, \epsilon^{-2} \right) ]$ (RL with 1 state, depth 1, and 2 actions). In Eq. 1, instead, the authors claim a complexity of $\widetilde{\mathcal{O}}(\epsilon^{-1})$, which is clearly impossible. It seems to me that the main problem is that the authors masked inside the $\widetilde{\mathcal{O}}$ instant dependent quantities such as $\Delta^{-2}$.

Secondly, I invite the authors to provide details on the last step behind the proof of Lemma B.1. To me, it seems that they upper-bound $P_\pi(\mathcal{E}) \le \epsilon$, but, then it is unclear how they upper bound the cumulative sum of rewards to $1$.

(minor) some assumptions in Lemma B.1 are not used. For instance, k-solvability seems to be not used within the proof.

Overall, I currently believe that all these issues could be solved, leading to results comparable (or maybe, slightly worse) to the one presented in Theorem 3.6. Nevertheless, the paper, at its current status, seems to be lacking in formal correctness.

**3. Weaknesses of the proposed algorithm (minor)**.

It has to highlighted that the algorithm proposed by the authors needs to be aware of the k parameter of K-QVI-solvability property. This parameter is often unknown in practice. I consider this to be a minor weakness, as the purpose of this work is not to propose an algorithm (with theoretical guarantees) that can be applied in practice, but rather it focuses on explaining why deep RL algorithms succed in practice.

**4. Minor comments:**
- Colors in Figure 2 seems to be swapped. I currently read the Figure as follows: PPO fails most likely with small values of k (however, I guess the opposite claim should be the correct one).

**Questions:**

See weakness section above.

---

> ### Author Response · Authors · 2023-11-16
> **Author response**
>
> We thank the reviewer for their detailed comments. We appreciate that they found our submission addresses an important problem and that it is "overall well-written and easy to understand." Below, we have responded to the weaknesses raised by the reviewer:
>
>  * **Novelty:** while our paper builds on Laidlaw et al. (2023), we argue that it is a significant contribution in its own right. Since the GORP algorithm introduced in the prior paper relies heavily on the environment being deterministic, it was not clear to us at first how to extend it to stochastic environments. While the novelty of our submission may seem "limited" in hindsight, we found that in discussions with other researchers and among ourselves, it was quite difficult to think of a stochastic analog of GORP. Furthermore, our theoretical analysis requires quite different tools than Laidlaw et al. They need to only analyze mean estimation-type problems for the deterministic case, which is quite simple. In contrast, we analyze the statistical generalization of complex function approximators across distributions of state-action pairs. Thus, we believe that our submission represents a significant advancement beyond prior work.
>  * **Correctness of theoretical claims:** we address the specific concerns of the reviewer regarding the theoretical claims of the submission below.
>     * *Sample complexity bound (Equation 1):* the instance-dependent $1/\Delta_k^2$ term is contained within the $A^{\bar{H}_k}$ term of the sample complexity bound presented in Equation 1, since $\bar{H}_k = k+ \log_A ( 1 / \Delta_k^2 )$ according to Definition 3.3. Thus, since the sample complexity bound is always higher than $1/\Delta_k^2$, it does not contradict lower bounds for the bandit case.
>     * *Last step of proof of Lemma B.1:* we do assume that the total rewards over an episode are bounded almost surely in $[0, 1]$, as we note in the second paragraph of Section 2 (page 3): "We assume that the total reward $\sum_{t=1}^T R_t(s_t, a_t)$
> is bounded almost surely in $[0, 1]$; any bounded reward function can be normalized to satisfy this assumption."
>     * *Assumptions in Lemma B.1:* we should have been more explicit about how $k$-QVI-solvability is used in the proof of Lemma B.1. It is used to show that $\pi^* \in \Pi(Q^k)$ is an optimal policy (by the definition of $k$-QVI-solvability), and thus that $J(\pi^*) = \max_\pi J(\pi)$. Thank you for pointing this out—we will make it explicit in the final version of the paper.
>
>    Overall, we do believe that any of these concerns affect the formal correctness of the paper. If the reviewer still believes there are issues with any of our theoretical claims, we invite them to let us know in a follow-up comment.
>  * **Weaknesses of the proposed algorithm:** we agree with the reviewer's assessment that, while SQIRL has hyperparameters that must be carefully tuned for a particular environment, the algorithm is still useful for theoretically understanding when and why deep RL works in practice. As we wrote at the top of page 9, "we do not claim that SQIRL is as practical as PPO or DQN, since it requires much more hyperparameter tuning; instead, we mainly see SQIRL as a tool for understanding deep RL."
>  * **Figure 2 colors:** the reviewer is correct that we accidentally swapped the colors in Figure 2. We have corrected the problem in an updated submission.

---

> > ### Comment · Reviewer_1dWJ · 2023-11-20
> > **Ack**
> >
> > I thank the authors for their in-depth rebuttals.
> >
> > Concerning the novelty, what I meant is the following aspect: the "explanation" of what is happening in deep RL methods takes large inspiration from an already existing work. In this sense, the novelty is limited. Notice that I ack. novelty in terms of extension to stochastic settings among the strength points.
> >
> > That being said, most of my theoretical concerns have been addressed. I have one final question for the authors. I see that the authors are operating under the assumption that cumulative sums are bounded in [0,1]. This assumption, however, is somehow different w.r.t. the one that is commonly adopted in the literature (i.e., rewards are bounded in [0,1] and cumulative returns in [0,T]). Can the authors modify their assumption to the standard setting? Is this possible? What kind of result would we obtain? In this way, the comparison in Table 1 can be more appreciated (as the other works mentioned, at least for the tabular setting, operate under this assumption).
> >
> > Minor comment:
> > - The authors refer to the work of Azar et al., 2017 in Table 1. It seems to me, however, that Azar 2017 builds algorithms and presents results for the regret metric.

---

> > > ### Author Response · Authors · 2023-11-21
> > > **Response to reviewer**
> > >
> > > We thank the reviewer for acknowledging the points in our rebuttal, and we are glad they appreciate the novelty of our extension to stochastic settings.
> > >
> > > With regards to the assumption on cumulative reward, we acknowledge that many other results in the literature make the assumption that per-timestep rewards are bounded in [0, 1], rather than the cumulative sum. Our results can be converted to this setting by scaling any reward function which is bounded as $R_t(s_t, a_t) \in [0, 1]$ by a factor of $1 / T$. There are then two quantities in our sample complexity bound (Equation 1) that need to be updated based on this scaled reward function:
> > >  1. The stochastic effective horizon depends on the $k$-gap $\Delta_k$, which scales with the reward function. However, to align with Laidlaw et al. (2023), we would also need to scale the numerator of the log term in the stochastic effective horizon definition (Definition 3.3) by $T$. Thus, the two $T$ terms cancel out and the stochastic effective horizon remains unchanged by scaling the reward function. See Theorem 5.4 of Laidlaw et al. for why the effective horizon is unchanged by scaling the reward function.
> > >  2. We also need to scale $\epsilon$ (the maximum suboptimality of the learned policy) based on the reward scale. In particular, we can write $T \epsilon_{[0, 1]} = \epsilon_{[0, T]}$, where $\epsilon_{[0, 1]}$ refers to the suboptimality when *cumulative* rewards are bounded in $[0, 1]$ and $\epsilon_{[0, T]}$ refers to the suboptimality when *per-timestep* rewards are bounded in $[0, 1]$ (making cumulative rewards bounded in $[0, T]$). Thus, $\epsilon_{[0, 1]} = \frac{1}{T} \epsilon_{[0, T]}$, meaning we need to add an additional $T$ factor to the top of (1).
> > >
> > > After scaling these two quantities, the sample complexity bound in (1) becomes
> > > $$\tilde{O}\left(k T^4 \alpha^{2(k-1)} A^{\bar{H}_k} D \log (\alpha D) / \epsilon \right)$$
> > > for the case when per-timestep rewards are bounded in $[0, 1]$ instead of the cumulative rewards. The only difference is an additional $T$ term.
> > >
> > > In Table 1 (the comparison of sample complexity bounds), we compare to bounds on the sample complexity under the same assumption that the *cumulative* rewards are bounded in [0, 1]. More details are listed below.
> > >  * **Tabular MDPs:** this sample complexity bound comes via the regret bound from Azar et al. (2017) of $\tilde{O}(\sqrt{H S A T})$ (using the notation that $H$ is the horizon) in the setting where per-timestep rewards are in [0, 1]. Jin et al. (2018) note that any regret bound of the form $\sqrt{C T}$ can be converted to a sample complexity bound of $O(H^2 C / \epsilon^2)$, so Azar et al.'s bound is equivalent to a sample complexity of $\tilde{O}(T^3 S A / \epsilon^2)$ (using our notation of $T$ being the horizon), still in the setting where per-timestep rewards are bounded in $[0, 1]$. Scaling $\epsilon$ by $T$ (since $T \epsilon_{[0, 1]} = \epsilon_{[0, T]}$) gives a bound of $\tilde{O}(T S A / \epsilon^2)$ in our setting where the cumulative rewards are bounded in [0, 1].
> > >  * **Linear MDPs:** this sample complexity bound is taken directly from Chen et al. (2022), who consider an identical setting to ours where cumulative rewards are bounded in [0, 1]. Thus, their sample complexity bound of $\tilde{O}(T^2 d^2 / \epsilon^2)$ is immediately applicable.
> > >
> > > In the initial submission we had scaled some of these bounds incorrectly, but we updated the paper and now believe they are completely correct. Thus, there is a fair comparison (identical reward scaling) between all bounds in Table 1.

---

> > > > ### Comment · Reviewer_1dWJ · 2023-11-21
> > > > **Ack**
> > > >
> > > > I thank the authors for their in-depth response. I have updated my score accordingly.
> > > > I have no further questions for the authors.

---

### Official Review · Reviewer_yWoc · 2023-10-31

**Soundness:** 3 good
**Presentation:** 3 good
**Contribution:** 2 fair
**Rating:** 6
**Confidence:** 3

**Summary:**

The paper extends the previous work GORP which studied the explanation of RL in deterministic environments by proposing and studying a new method under stochastic environments under function approximation. GORP shows that the success of both PPO and DQN can simply be explained by a simple procedure of greedily improving a few steps over a value function of random policy. SQIRL extends the algorithm and analysis for stochastic environments. The results demonstrate that the performance of the proposed method SQIRL correlates with the performance of PPO and DQN in stochastic environments.

**Strengths:**

1. The paper addresses the key limitation of prior work: GORP which showed that performance of deepRL can be explained by just acting greedily with respect to value function of a random policy in deterministic environments. GORP does not extend to stochastic environments directly and the authors propose one way to extend the algorithm to stochastic environments by using function approximation.
2. The authors theoretically explain the sample complexity of their proposed algorithm in terms of psuedo-dimension of the function class along with a combination of concentration and FQI analysis.  The authors also demonstrate that this sample complexity is closely related to sample complexity of Deep RL algorithms used in practice.
3. The method SQIRL is tested extensively over a set of 150 environments from previous work GORP. The environments are made stochastic by using sticky actions and show that whenever PPO or DQN does well, SQIRL does well 78% of the time too.

**Weaknesses:**

1. My major concern is around the novelty of the proposed approach:
a. To address stochasticity an open loop trajectory optimization is replaced by FQI. This is not new in my opinion:
Prior works:
[1] Considers a tree search with the FQI procedure along with analysis to account for stochastic environments.
[2] Considers a FQI analysis of H-step lookahead under empirical distributions for a more general setting of learned models (A.3)
[3] Considers a similar FQI setting with learned models where exploratory policy is the dataset policy
A proper discussion on the novelty of using FQI to replace open-loop trajectory optimization along with a comparison with these prior works is warranted.
2. The strong claim of in-distribution generalization: On page 6 the authors claim that if we can properly regress for approximate-Q then we can also estimate the maximum action properly. The claim seems to be too strong without evidence: With finite data, the Q will almost always have errors. These errors will lead to perhaps suggesting actions OOD and lead to overestimation bias commonly observed in deep RL. This analysis seems to be missing in analysis of FQI for SQIRL.
3. A minor nitpick: Unlike GORP, the whole set of stochastic environments are not explained by SQIRL - I think the title and introduction implying something more stronger that it should?
4. Sticky action is a particular kind of stochasticity - More ways of inducing stochasticity and varying the noise std can make the empirical experiments stronger.
[1]: https://arxiv.org/pdf/2107.01715.pdf
[2]: https://arxiv.org/pdf/2107.01715.pdf
[3]: https://arxiv.org/pdf/2008.05556.pdf

**Questions:**

None

---

> ### Author Response · Authors · 2023-11-16
> **Author response**
>
> We thank the reviewer for their insightful comments. We are glad they found that our paper “addresses the key limitation of prior work” and that our experiments are “extensive.” Below we have responded to the weaknesses raised in their review.
>
>  * **Novelty compared to previous work:** we acknowledge that tree search and FQI have been extensively used and analyzed in prior works, including those the reviewer cited. However, the papers mentioned in the review use a model for planning. Unlike these works, our focus is on learning an optimal policy within a model-free setting, where no access to a pre-existing model is available. This introduces a higher level of complexity as we are constrained to sample episodes from a fixed initial state distribution and cannot reset to arbitrary states for further planning. This means that it is not possible to apply standard tree search procedures—instead, one of our key contributions is constructing an RL algorithm, SQIRL, which behaves in some ways like tree search but is actually completely model-free. We will clarify this point and compare to the work cited by the reviewer in the final version of the paper if accepted.
>  * **Claim of in-distribution generalization:** the reviewer notes that in many Q-learning-like RL algorithms, the Q values of states which are rarely seen or actions which are rarely taken or may be significantly mis-estimated. However, our theoretical analysis shows that this is *provably* not the case for SQIRL. Since SQIRL explores by taking actions uniformly at random before running FQI, no actions are sampled more often than others in expectation, meaning that no actions are any more "out-of-distribution" than others. According to Lemma B.2, this means that the mean-squared error of the Q-function can blow up by a factor of $A$ (the number of actions) at each step of FQI, but no more. We then combine this error bound with Markov's inequality in Lemma B.3 to show that the probability of any action's Q-value being off by too much is bounded. This allows us to give the bounds on sample complexity of SQIRL in Theorem 3.6.
>  * **Whole set of stochastic environments is not explained by SQIRL:** we will revise our language in the introduction to indicate that we do not provide a complete explanation for deep RL performance in all possible environments. Instead, our analysis is aimed at generally understanding when and why deep RL works across a broad range of environments. We note that in Laidlaw et al. (2023), GORP also does not solve all the deterministic BRIDGE environments—is there another way in which you are claiming that GORP "explains" all the environments?
>  * **Other types of stochasticity:** we agree that there are other ways of adding stochasticity to environments besides sticky actions. We primarily focus on sticky actions based on the evidence provided by Machado et al. [1], who explore a number of ways of inducing stochasticity in Atari games. They conclude that sticky actions are the best way to introduce stochastic transitions:
>
>    > Sticky actions [leverages] some of the main benefits of other [stochasticity] approaches without most of their drawbacks. It is free from researcher bias, it does not interfere with agent action selection, and it discourages agents from relying on memorization. The new environment is stochastic for the whole episode, generated results are reproducible, and our approach interacts naturally with frame skipping and discounting.
>
>    Thus, following their conclusions, we decided to focus on creating stochastic environments using sticky actions.
>
> [1] Machado et al. Revisiting the arcade learning environment: Evaluation protocols and open problems for general agents. Journal of Artificial Intelligence Research, 2018.

---

> > ### Comment · Reviewer_yWoc · 2023-11-18
> > **Reviewer response**
> >
> > Thank you for the rebuttal.
> >
> > My concerns still remain about novelty despite the model-free/model-based argument since any H-step trajectory optimization can be replaced by H-step QVI. This has been used in prior works before including Laidlaw et al. (2023). I think my concerns about in-distribution generalization are addressed.
> >
> > **is there another way in which you are claiming that GORP "explains" all the environments?**
> >
> > Citing this line from their work:
> > >  This proportion rises to four-fifths among environments that PPO [9], a popular deep RL algorithm, can solve efficiently (Table 1). Conversely, when this property does not hold, PPO is more likely to fail than succeed—and when it does succeed, so does simply applying a few steps of lookahead on the Q-function of the random policy
> >
> > Finally, I think ICLR allows paper revisions. It would also help to see the promised changes updated in the paper. eg "we will revise our language in the introduction to indicate that we do not provide a complete explanation for deep RL performance in all possible environments." and "Novelty compared to previous work:"

---

> > > ### Author Response · Authors · 2023-11-21
> > > **Response to reviewer**
> > >
> > > Thank you for responding to our rebuttal.
> > >
> > > **Replacing trajectory optimization via QVI:** While it is somewhat straightforward to replace $k$-step trajectory optimization with $k$-step QVI/FQI, they are only equivalent in the deterministic setting. This is because in the stochastic setting one needs to learn a closed-loop *policy*, rather than an open-loop *sequence of actions*. Thus, $k$-step trajectory optimization will not work in stochastic environments. This means that our analysis requires quite different tools from Laidlaw et al. Their theoretical approach is focused on bounding the error of estimating the multi-step Q-functions for each $k$-step action sequence starting from a single state, which is a simple mean estimation problem. In contrast, we focus on learning a *policy* which must perform well in expectation under a distribution over potentially infinitely many states. This requires analyzing how FQI propagates errors through *distributions* over states and transitions, and also how it can enable *generalization* to new states unseen during training. Neither of these aspects are considered by Laidlaw et al. Thus, while our SQIRL algorithm is nearly equivalent to their GORP algorithm in the *deterministic* setting, SQIRL is much more complex to anlayze in the stochastic setting. Our analysis therefore represents a significant advance beyond prior work.
> > >
> > > **Explaining all the environments:** We find a similar result to the quoted one from Laidlaw et al. when comparing which environments are solved by SQIRL and which are solved by PPO/DQN. In the table below, we cross-tabulate the number of environments in which SQIRL succeeds or fails at learning an optimal policy with the numbers in which PPO/DQN succeed or fail. We see that when SQIRL suceeds—suggesting that our theoretical assumptions are met—then PPO succeeds 97% of the time and DQN 82% of the time. When SQIRL fails, then PPO succeeds only 41% of the time and DQN only 24% of the time. Thus, we believe SQIRL explains a good deal of the performance of PPO and DQN in stochastic environments.
> > >
> > >    |              | SQIRL succeeds | SQIRL fails |
> > >    |--------------|----------------|-------------|
> > >    | PPO succeeds | 66             | 36          |
> > >    | PPO fails    | 2              | 51          |
> > >    | DQN succeeds | 56             | 21          |
> > >    | DQN fails    | 12             | 66          |
> > >
> > > We note that the results of Laidlaw et al. (2023) also do not perfectly explain all of the environments. Table 2 of their paper shows that their sample complexity bounds are only 86% accurate at predicting whether PPO and DQN will succeed or fail. Furthermore, the median ratio between the sample complexity of their GORP algorithm and the performance of PPO is 7.3 and for DQN it is 11. In contrast, the median ratio between SQIRL and PPO in our experiments is 1.0 and for DQN it is 0.55, suggesting that in some ways SQIRL is a *better* explanation than GORP since its sample complexity is much closer to that of deep RL algorithms.
> > >
> > > **Updating the paper:** We have updated some of the language in the introduction to make it clear that we do not claim we can explain deep RL's performance in all environments. In the original submission we had already noted that
> > >
> > > > There are still some environments in our experiments where SQIRL fails while PPO or DQN succeeds, suggesting lines of inquiry for future research to address.
> > >
> > > We also updated the related work to include the papers mentioned in the review and clarify the novelty of our analysis compared to them.

---

> > > > ### Comment · Reviewer_yWoc · 2023-11-23
> > > >
> > > > Thank you for the detailed clarifications. While I still have reservations about the novelty of QVI to replace TrajOpt, I find the rebuttal satisfactory. I have raised my score.

---

### Official Review · Reviewer_kbuB · 2023-11-02

**Soundness:** 3 good
**Presentation:** 3 good
**Contribution:** 2 fair
**Rating:** 5
**Confidence:** 2

**Summary:**

This paper considers a setting where exploration is not needed. They formalize the setting in Def 3.1. They also design an algorithm and show that the algorithm is sample-efficient in the setting they consider. They also conduct experiment to validate their idea.

**Strengths:**

The setting they consider is closely related to the tasks in the application.

**Weaknesses:**

1. The theory is simple and straightforward. In fact, in the setting they considered, exploration is not needed.

2. It is unclear whether their algorithm can be adapted to the scenario where exploration is needed.

**Questions:**

1. See the 'Weakness' section.

2. Figure 2 shows that PPO fails in most tasks with k=1, which contradicts the claim 'Furthermore, these are the environments where deep RL algorithms like PPO are most likely to find an optimal policy' on Page 5. Can you provide an explanation?

3. Apart from the tasks in Bridge, can you verify Def 3.1 for other tasks, including go and robotic?

---

> ### Author Response · Authors · 2023-11-16
> **Author response**
>
> We thank the reviewer for their comments. We have addressed the weaknesses and questions they mentioned below:
>
>  * **”The theory is simple and straightforward… [since] exploration is not needed.”** Just to clarify, in *any* RL problem exploration is needed. However, in the setting we consider, random exploration seems to be enough, i.e., strategic exploration is not necessary. One of our central points in the paper is that many common RL benchmarks *can* be solved with random exploration, as shown by the success of deep RL algorithms that explore randomly. Our contribution is to explain—both theoretically and empirically—how random exploration works well in practice with complex function approximators like neural networks, since existing theory suggests that random exploration can result in exponentially bad sample complexity.
>
>    Furthermore, we argue that our theory is not simple or straightforward, since very few previous works in the RL theory literature have been able to explain why random exploration is sample efficient in practice despite being exponentially inefficient in the worst case. The few papers that have focused on random exploration either rely on assumptions that do not seem to hold in realistic environments or they do not consider generalization with complex function approximators. In contrast, our theory both rests on assumptions that are verified in common RL benchmarks and is very general, as it can be applied to even quite complex function approximators like neural networks.
>  * **”Figure 2 shows that PPO fails in most tasks with k=1.”** This was actually a typo on our part; the labels should have been reversed. We have updated the figure accordingly and it now correctly shows that PPO succeeds in most of the MDPs with k = 1.
>
>  * **”Apart from the tasks in Bridge, can you verify Def 3.1 for other tasks, including go and robotic?”** In this paper, our focus is on model-free RL in environments with discrete action spaces. Model-free RL does not work very well for Chess or Go—the best RL algorithms for solving these games have consistently been model-based, like AlphaZero and MuZero. Furthermore, they are two-player games, while we consider single-agent environments. Most robotics tasks are also outside of the scope of our analysis since they have continuous action spaces. While we hope to extend our work from discrete actions to continuous action spaces in the future, our current submission already covers a large number of common RL benchmarks, like Atari games, Procgen, and MiniGrid. Thus, we believe that verifying Definition 3.1 ($k$-QVI-solvability) in these environments is already enough to show that this property holds in a large proportion of the types of environments where model-free RL works in practice.

---

### Meta-Review · Area_Chair_zUJS · 2023-12-11

**Metareview:**

This paper aims to further the understanding of when existing deep RL methods will be successful. Previous work has shown that the algorithms could perform well in deterministic environments after a few steps of policy iteration. This paper explains that argument in environments with stochastic transition dynamics. It presents a theoretical analysis of an algorithm to perform policy iteration like updates and experimental evidence that when this algorithm performs well, so does the standard deep RL algorithms. Although this does not fully explain when or why deep RL methods will work well, it expands on the current understanding. I recommend this paper for acceptance.

**Justification For Why Not Higher Score:**

The main reason this paper should not be given an oral presentation is because it is an extension of a previous result to a new setting. While the result is nontrivial and noteworthy, I think it deserves a spotlight, but perhaps not an oral presentation.

**Justification For Why Not Lower Score:**

This paper could be given a poster, but I think giving it a spotlight presentation will increase awareness of the work, leading others to ask other interesting questions about when deep RL will be successful.

---

### Decision · Program_Chairs · 2024-01-16

Accept (spotlight)